# The structural basis of lipid scrambling and inactivation in the endoplasmic reticulum scramblase TMEM16K

Simon R. Bushell [1,17], Ashley C.W. Pike [1,17], Maria E. Falzone[2,17], Nils J.G. Rorsman[3,10], Chau M. Ta[3,11], Robin A. Corey[4], Thomas D. Newport[4,12], John C. Christianson [5], Lara F. Scofano [3], Chitra A. Shintre [1,13], Annamaria Tessitore[1,14], Amy Chu[1,15], Qinrui Wang[1,4], Leela Shrestha[1], Shubhashish M.M. Mukhopadhyay [1], James D. Love[6,16], Nicola A. Burgess-Brown[1], Rebecca Sitsapesan[3,18], Phillip J. Stansfeld [4,18], Juha T. Huiskonen [7,18], Paolo Tammaro[3,18], Alessio Accardi [2,8,9,18] & Elisabeth P. Carpenter [1,18]

Membranes in cells have defined distributions of lipids in each leaflet, controlled by lipid scramblases and flip/floppases. However, for some intracellular membranes such as the endoplasmic reticulum (ER) the scramblases have not been identified. Members of the TMEM16 family have either lipid scramblase or chloride channel activity. Although TMEM16K is widely distributed and associated with the neurological disorder autosomal recessive spinocerebellar ataxia type 10 (SCAR10), its location in cells, function and structure are largely uncharacterised. Here we show that TMEM16K is an ER-resident lipid scramblase with a requirement for short chain lipids and calcium for robust activity. Crystal structures of TMEM16K show a scramblase fold, with an open lipid transporting groove. Additional cryo-EM structures reveal extensive conformational changes from the cytoplasmic to the ER side of the membrane, giving a state with a closed lipid permeation pathway. Molecular dynamics simulations showed that the open-groove conformation is necessary for scramblase activity.

[1] Structural Genomics Consortium, Nuffield Department of Medicine, University of Oxford, Old Road Campus Research Building, Roosevelt Drive, Oxford OX3 7DQ, UK. [2] Department of Biochemistry, Weill Cornell Medical School, 1300 York Avenue, New York, NY 10065, USA. [3] Department of Pharmacology, University of Oxford, Mansfield Road, Oxford OX1 3QT, UK. [4] Department of Biochemistry, University of Oxford, South Parks Road, Oxford OX1 3QT, UK. [5] Nuffield Department of Rheumatology, Orthopaedics and Musculoskeletal Sciences, University of Oxford, Windmill Road, Oxford OX3 7LD, UK. [6] Department of Biochemistry, Albert Einstein College of Medicine, 1300 Morris Park Avenue, Bronx, NY 10461-1602, USA. [7] Division of Structural Biology, Wellcome Centre for Human Genetics, University of Oxford, Roosevelt Drive, Oxford OX3 7BN, UK. [8] Department of Anesthesiology, Weill Cornell Medical School, 25 East 68th Street, New York, NY 10065, USA. [9] Department of Physiology and Biophysics, Weill Cornell Medical School, 1300 York Avenue, New York, NY 10065, USA. [10] Present address: OxSyBio, Atlas Building, Harwell Campus, Didcot, Oxfordshire OX11 0QX, UK. [11] Present address: Department of Cardiology, Washington University in St. Louis, St. Louis, MO 63110, USA. [12] Present address: Oxford Nanopore Technologies, Oxford Science Park, Oxford OX4 4DQ, UK. [13] Present address: Vertex Pharmaceuticals Ltd, Milton Park, Oxfordshire OX14 4RW, UK. [14] Present address: Nuffield Division of Clinical Laboratory Sciences, Oxford University, Oxford OX3 9DU, UK. [15] Present address: Department of Biochemistry, Oxford University, Oxford OX1 3QT, UK. [16] Present address: Novo Nordisk A/S, Novo Nordisk Park, 2760 Måløv, Denmark. [17] These authors contributed equally: Simon R. Bushell, Ashley C.W. Pike, Maria E. Falzone. [18] These authors jointly supervised this work: Rebecca Sitsapesan, Phillip J. Stansfeld, Juha T. Huiskonen, Paolo Tammaro, Alessio Accardi, Elisabeth P. Carpenter. Correspondence and requests for materials should be addressed to E.P.C. (email: liz.carpenter@sgc.ox.ac.uk)

Cells and their organelles are enclosed by lipid bilayers and the lipid composition of either side of these membranes is controlled by active transporters (flippases and floppases) and passive scramblases, which equilibrate lipids between the membrane leaflets[1]. Many lipids are synthesised on the cytoplasmic side of the endoplasmic reticulum (ER) membrane which, unlike the plasma membrane (PM), has a symmetrical lipid distribution, suggesting a role for scramblases in the ER. To date, specific ER scramblases have not been identified and characterised[2]. The ten members of the TMEM16 scramblase/channel family of integral membrane proteins (also known as anoctamins ANO1-10) show a surprising diversity of function, being either $Ca^{2+}$-activated chloride channels (TMEM16A and B)[3–5] or $Ca^{2+}$-activated lipid scramblases with non-selective ion channel activity (TMEM16C, D, F, G and J)[6–9]. Although mammals have ten members in this family, lower eukaryotes generally have fewer TMEM16s. While some members of the family (TMEM16A, B, F) reside in the plasma membrane, others, including TMEM16K[10] may function in intracellular membranes. TMEM16K is a widely distributed[11], but relatively unstudied member of the TMEM16 family. Truncations and missense variants of TMEM16K (ANO10) are associated with the autosomal recessive spinocerebellar ataxia SCAR10[12,13] (as known as ARCA3[14–16] or ATX-ANO10[17]). SCAR10 causes cerebellar ataxia, with cerebellar atrophy evident in magnetic resonance imaging scans of brains and coenzyme Q10 deficiency found in muscle biopsy, fibroblasts and cerebrospinal fluid[12,13,18,19]. Some patients also have epilepsy and cognitive impairment[14,15]. Knockout studies in Drosophila[20] and mice[21] have suggested that loss of TMEM16K homologue function affects spindle formation[20], $Ca^{2+}$ signalling[21] and apoptosis[20,21].

Structural studies have gone some way towards explaining how TMEM16 family members function as channels or lipid scramblases. The crystal structure of the *Nectria haematococca* TMEM16 (nhTMEM16), a fungal lipid scramblase with non-selective channel activity, revealed a dimer arranged in a bi-lobal 'butterfly' fold, with each subunit containing a two $Ca^{2+}$ ion binding site and ten transmembrane (TM) helices[22]. Each monomer has a hydrophilic, membrane-spanning groove that provides a route for lipid headgroups to move across membranes. Molecular dynamics (MD) simulations subsequently confirmed this lipid scrambling mechanism in silico[23,24]. Structures of the mouse TMEM16A chloride channel revealed an alternative conformation, with two groove-associated transmembrane α-helices blocking the top of the scramblase groove, forming a closed pore[25–27]. In addition, while this paper was under review, structures of the fungal homologues afTMEM16[28] and nhTMEM16[29] and the mouse TMEM16F[30] were published, showing a range of conformations for the fungal homologues, and closed confirmations of mTMEM16F, involving small movements of helices near the groove.

In spite of its patho-physiological relevance, TMEM16K remains a poorly characterised member of the TMEM16 family, as its cellular localisation, function, regulation and structure are largely uncharacterised. Here we show that TMEM16K is an ER-resident lipid scramblase with non-specific ion channel activity and a dependence on calcium ions and short chain lipids for optimum activity. We present structures of TMEM16K solved by both X-ray crystallography and cryo-electron microscopy, revealing a classic scramblase fold[22], with extensive conformational changes propagated from the cytoplasmic to the ER face of the membrane, which lead to opening or closing of the lipid transporting groove. In particular, these structures reveal the range of conformations available for scrambling by a mammalian scramblase. We observe both changes that do not rely on changes in $Ca^{2+}$-ion binding and additional, smaller changes that occur when $Ca^{2+}$ ions are removed. We use MD simulations to confirm that in TMEM16K the open groove conformation is necessary for scramblase activity.

## Results

**TMEM16K is an ER resident lipid scramblase**. The identity of the membrane environments in which TMEM16K resides has not been clearly established[11,21,31]. To investigate this question, we assessed the subcellular localisation of TMEM16K (initially using a human TMEM16K construct with a TEV-His$_{10}$-FLAG tag, including a tobacco etch virus (TEV) protease cleavage site) heterologously expressed in adherent monkey kidney fibroblasts (COS-7) cells. We observed significant co-localisation with ER membranes stained for either the ER-resident chaperone calnexin (CNX, Fig. 1a, b) or the ER ubiquitin ligase Hrd1 (Supplementary Fig. 1a, b, Supplementary Table 1). This observation was supported by staining of endogenous TMEM16K in human bone osteosarcoma epithelial (U2OS) cells, which also co-localised with the ER marker KDEL (Fig. 1c, d, Supplementary Table 1). Together, these data are consistent with TMEM16K primarily residing in the ER membrane. We also considered the possibility that a small fraction of TMEM16K might localise to the PM, where it could function as a $Ca^{2+}$-dependent ion channel. To test this, we measured the whole-cell currents in human embryonic kidney 293T (HEK-293T) cells expressing TMEM16K in the presence of either 300 nM or 78 μM intracellular free $Ca^{2+}$ concentration ($[Ca^{2+}]_i$). These $[Ca^{2+}]_i$ are sufficient to activate two TMEM16K homologues, TMEM16A and TMEM16F, a PM chloride channel[3–5,7] and a lipid scramblase with non-selective ion channel activity[6,32], respectively. However, the currents detected in TMEM16K-transfected cells did not differ from those seen in mock-transfected cells in either condition (Fig. 1e, f), suggesting that TMEM16K is not trafficked to PM and/or does not mediate PM $Ca^{2+}$-activated currents. These results are in agreement with reports suggesting that no scramblase activity was detected at the PM when TMEM16K was expressed in TMEM16F$^{-/-}$ mouse cells[11].

Next, we used a dithionite-based scrambling assay[33] (Fig. 2a) to investigate whether purified TMEM16K has $Ca^{2+}$-dependent scramblase activity, as seen in mammalian TMEM16F[6,7,34] and the fungal afTMEM16[33] and nhTMEM16[22] homologues. We found that purified hTMEM16K, reconstituted into lipid vesicles, formed from a 7:3 mixture of phosphatidylcholine (PC) and phosphatidylglycerol (PG), with 16:0–18:1C acyl tails, transports lipids with a scrambling rate constant of ~0.003 s$^{-1}$ in 0.5 or 2 mM $Ca^{2+}$, corresponding to ~100 lipids translocated s$^{-1}$ (Fig. 2b, d, Supplementary Fig. 3a). Without $Ca^{2+}$, the rate constant decreased ~4-fold, indicating that TMEM16K scramblase activity is $Ca^{2+}$ regulated. Other TMEM16 scramblases, such as afTMEM16, are faster[34–36], ~20-fold in 0.5 mM $Ca^{2+}$ and ~5-fold without $Ca^{2+}$ (Fig. 2b, d). To test if the slow rate of scrambling by TMEM16K is dependent on the expression system used, we compared the activity of TMEM16K purified from insect-derived Sf9 cells (Fig. 2d) and from mammalian HEK-293 cells (Supplementary Fig. 3a), and found they have comparable properties (Fig. 2d). Thus, the slow rates of scrambling in 16:0–18:1 lipids reflect an intrinsic property of purified TMEM16K.

We recently showed that the activity of the fungal afTMEM16 scramblase is modulated by membrane thickness, with thicker membranes inhibiting activity[37]. Because TMEM16K resides in the ER membrane, which is thinner than the plasma membrane[38,39], we hypothesised that its scrambling activity might be enhanced in membranes formed from lipids with shorter acyl tails[40]. We found that protein-free liposomes formed

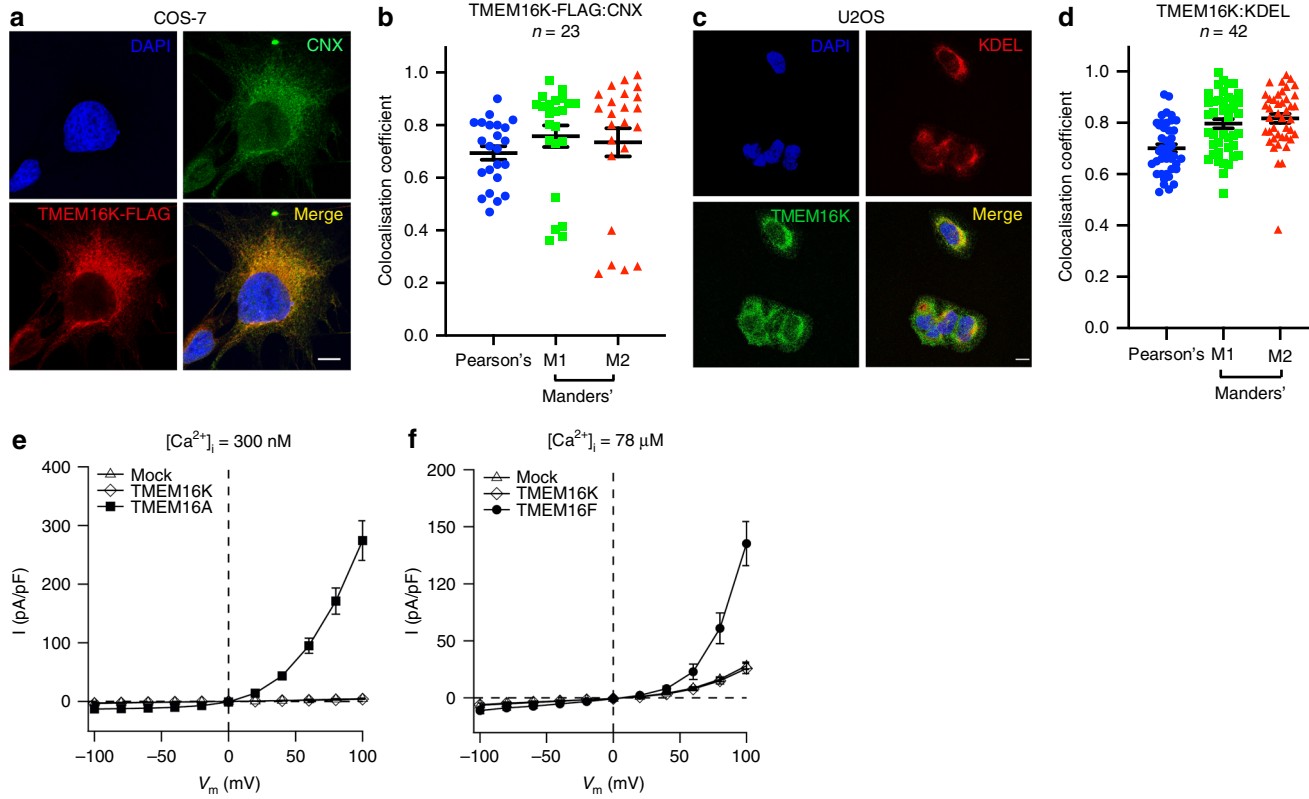

**Fig. 1** TMEM16K localises predominantly to the ER. **a** Representative confocal images of COS-7 cells expressing TMEM16K-TEV-His$_{10}$-FLAG. Cells were stained for TMEM16K (anti-FLAG: red), the ER resident protein calnexin (CNX: green) and nuclei (DAPI: blue). In the merged panel (lower right), the degree of TMEM16K and CNX overlap is shown (yellow). Scale bars = 10 μm; magnification: ×63. **b** Pearson's correlation and Manders' (M1 and M2) overlap coefficients determined for TMEM16K (FLAG) and CNX. Mean and s.e.m. are shown, $n = 23$ cells. **c** Representative confocal images of U2OS cells stained using antibodies to endogenous TMEM16K (green), the ER resident protein signature KDEL (red), along with DAPI staining of nuclei and the merge. Scale bars = 10 μm; magnification: ×63. **d** Quantitative analysis of TMEM16K and KDEL colocalisation in U2OS cells (as in **b**). Mean and s.e.m. are shown, $n = 42$. **e** Whole-cell current versus voltage relationships for mock-transfected HEK-293T cells (Mock, $n = 15$) or cells expressing TMEM16A ($n = 13$) or TMEM16K ($n = 13$) are represented as mean ± s.e.m. [Ca$^{2+}$]$_i$ was 300 nM. **f** Whole-cell current versus voltage relationships for mock-transfected HEK-293T cells (Mock, $n = 12$) or cells expressing TMEM16F ($n = 17$) or TMEM16K ($n = 11$) are represented as mean ± s.e.m. [Ca$^{2+}$]$_i$ was 78 μM. Source data are provided as a Source Data file

from 100% short chain 14:0–14:0 PC and PG lipids were not stable, whereas vesicles formed from a 50–50 mixture of short 14:0–14:0 and long 16:0–18:1 lipids were stable. When TMEM16K is reconstituted in these mixed membranes, its scrambling activity dramatically increases (Fig. 2c, d, Supplementary Fig. 3b); in the presence of 0.5 mM Ca$^{2+}$ the rate constant of scrambling is ~0.01 s$^{-1}$, a tenfold increase compared to that seen in the long chain lipid membranes, and is comparable to that of afTMEM16 (Fig. 2d). In the absence of Ca$^{2+}$ the rate constant of scrambling is ~0.001 s$^{-1}$, again comparable to that of afTMEM16. Thus, the activity of TMEM16K increases in thinner membranes, suggesting that the membrane thickness is a general modulator of TMEM16 scramblase activity.

Next, we tested whether the TMEM16K scramblase displays a specific lipid selectivity using different nitrobenzoxadiazole (NBD)-labelled probes in the background of the thinner membranes and found that TMEM16K scrambles phosphatidylethanolamine (PE) and phosphatidylcholine (PC) equally well, with rate constants of ~0.01 s$^{-1}$ in the presence of 0.5 mM Ca$^{2+}$ (Fig. 2e, f, Supplementary Fig. 3c, d). Surprisingly, TMEM16K scrambles phosphatidylserine (PS) ~3-fold slower, with a rate constant of ~0.0035 s$^{-1}$ in the presence of 0.5 mM Ca$^{2+}$. Therefore, TMEM16K, unlike its fungal homologues afTMEM16 and nhTMEM16[22,33], is at least moderately selective among

different phospholipids. This observation is also consistent with the relatively low content of PS in the ER compared to the PM.

We used an end-point ion flux assay to test if TMEM16K also functions as a non-selective ion channel by determining whether a reconstituted protein dissipates a KCl gradient in the absence of ionophores[8,33]. The assay measures the fraction of liposomes that contain at least one active non-selective channel by determining the loss of Cl$^-$ content from the proteoliposomes in the presence of a salt gradient (Fig. 2g). We found that TMEM16K functions as a non-selective channel in mixed chain length liposomes, with ~58% vesicles containing at least one active TMEM16K channel (Fig. 2h). The activity of TMEM16K is minimally Ca$^{2+}$-dependent as the fraction of channel-active liposomes in the absence of Ca$^{2+}$ ions drops to ~43% (Fig. 2h). This could either reflect the relatively high constitutive activity of TMEM16K in the absence of Ca$^{2+}$ seen in the scrambling assay or be due to differential regulation of the channel and scramblase activities by Ca$^{2+}$. Further work will be needed to elucidate this.

In summary, we have shown that TMEM16K is a lipid scramblase with non-selective ion channel activity that appears to be mainly resident in the ER membrane. Removal of Ca$^{2+}$ significantly decreases the lipid scramblase activity. This activity is also sensitive to the lipid chain length of the surrounding membrane and displays moderate selectivity between different

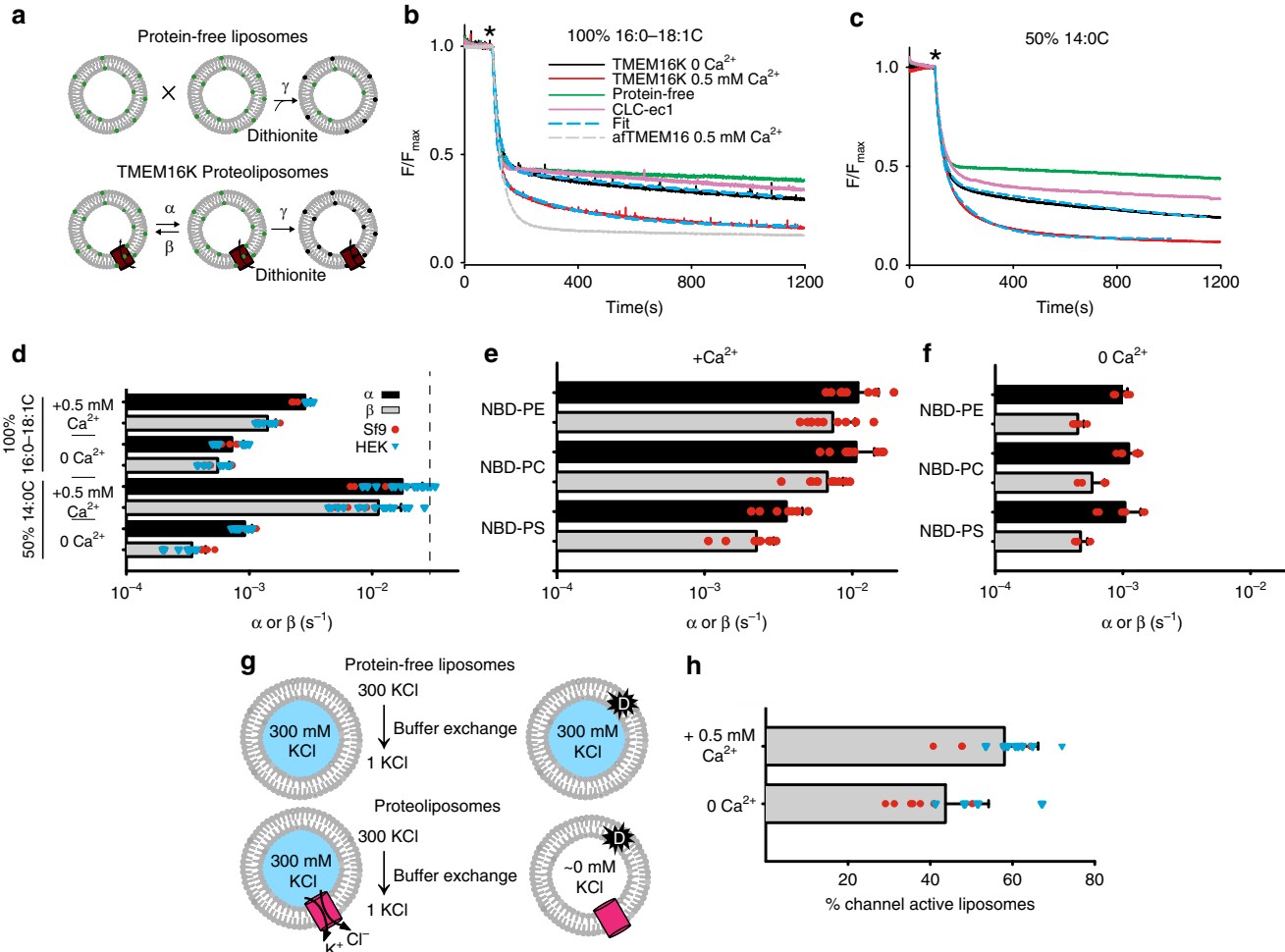

**Fig. 2** Characterisation of TMEM16K Ca$^{2+}$ activated scramblase activity. **a** Schematic for the dithionite-based phospholipid scramblase activity. **b**, **c** Representative time courses of the dithionite-induced fluorescence decay in liposomes made from **b** 100% 16:0–18:1 C acyl chains or **c** 50% 14:0 C acyl chains with protein-free liposomes (green), TMEM16K proteoliposomes with 0.5 mM (red) or without (black) Ca$^{2+}$, or in proteoliposomes containing the CLC-ec1 H$^+$/Cl$^-$ exchanger that does not scramble (pink). Cyan dashed lines represent the fit to Eq. 1, the analytical solution of the scheme in (**a**)[36]. * denotes addition of dithionite. Traces shown here are from Sf9-expressed TMEM16K. **d** Scrambling forward ($\alpha$) and reverse ($\beta$) rate constants for TMEM16K at 0.5 mM, and 0 mM Ca$^{2+}$ comparing liposomes composed of 100% 16:0–18:1 C acyl chains (top bars) or 50% 14:0 C acyl chains (bottom bars). Individual rate constants are shown as red circles (from Sf9-expressed TMEM16K) or blue triangles (from HEK-expressed TMEM16K). The dashed line indicates the rate for afTMEM16 in 0.5 mM Ca$^{2+}$. **e**, **f** Scrambling forward ($\alpha$) and reverse ($\beta$) rate constants for TMEM16K at 0.5 mM (**a**), and 0 mM Ca$^{2+}$ (**b**) in liposomes composed of 50% 14:0 C acyl chains with different NBD-labelled lipids (NBD-PE, PC, or PS). Individual rate constants are shown as red circles. All data in (**e**) and (**f**) are from Sf9-expressed wild-type TMEM16K. All rate constants are reported as the mean ± SD. **g** Schematic for the non-selective channel activity assay. **h** Fraction of the liposomes containing at least one active TMEM16K channel in the presence of 0.5 mM Ca$^{2+}$ or without Ca$^{2+}$ as red circles (from Sf9-expressed TMEM16K) or blue triangles (from HEK-expressed TMEM16K). All data are reported as the mean ± SD. Source data are provided as a Source Data file

headgroups of the lipid substrate. We propose that TMEM16K could contribute to the background scramblase activity required to maintain the even distribution of lipids on either side of the ER membrane. This activity is particularly important in the ER as lipids are synthesised on the cytoplasmic face of the membrane, so constant scramblase activity would be required to maintain the even distribution.

**Crystal structure determination and overall architecture**. To understand the structural basis of lipid scrambling in TMEM16K, we initially solved its structure by X-ray crystallography. We obtained preliminary diffraction data from vapour diffusion (VD) crystals of full-length human TMEM16K, which we improved by lipidic cubic phase (LCP) crystallisation to obtain 3.4 Å resolution data. Both structures were solved by molecular replacement (Methods section, Supplementary Table 2). The structures

were similar (monomer all atom root mean square deviation (r.m.s.d.) = 0.75 Å), although the crystal packing differs (Supplementary Fig. 4a–c), confirming that the conformation is not dictated by crystal contacts.

hTMEM16K is a symmetrical homodimer that adopts the classic TMEM16 butterfly fold[22,25–27] where each monomer comprises an N-terminal cytoplasmic domain (NCD) followed by a ten TM domain (TM1−TM10) and a C-terminal cytoplasmic α10 (Fig. 3a, b). The ER luminal surface of hTMEM16K is relatively compact, lacking the long, disulphide-bonded loops seen in mTMEM16A (Fig. 3a, b, Supplementary Figs. 2, 4e). The N-terminal, cytoplasmic domain of hTMEM16K has a four-stranded β sheet (β1, β2, β3 and β8), with three helices on one side (α1, α3 and α4). These helices are covered by the β6−β7 hairpin insertion, which is unique to TMEM16K (Fig. 3c, d, Supplementary Fig. 5a). This β6−β7

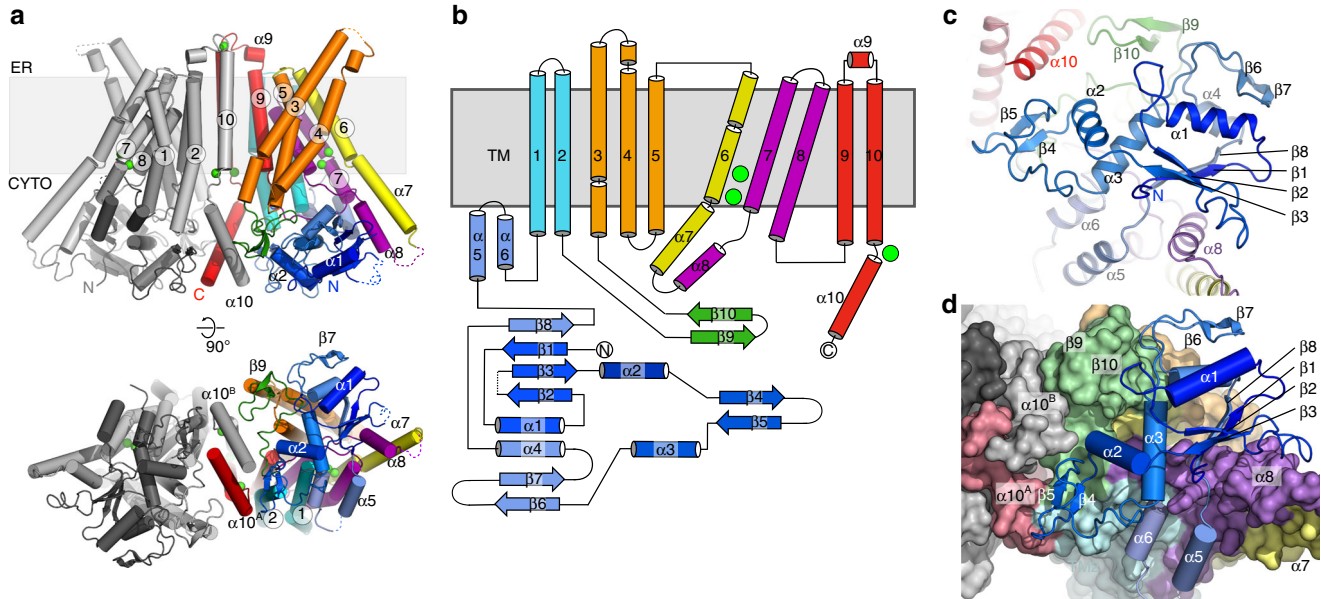

**Fig. 3** TMEM16K fold and cytoplasmic domain in the lipid scramblase conformation. **a** The structure of hTMEM16K viewed from the membrane plane and cytoplasm. In chain A the secondary structural elements are coloured as follows: the cytoplasmic domain (blue), TM1 and 2 (cyan), the β9−β10 hairpin (green), TM3, 4, 5 (orange), TM6 and α7 (yellow), α8, TM7 and 8 (purple), TM9, TM10 and α10 (red). Chain B is coloured grey. $Ca^{2+}$ are coloured bright green. **b** TMEM16K secondary structure elements, coloured according to (**a**). **c** Cytoplasmic domain (blue), α7 and α8 (yellow/purple) and α10 (red) of TMEM16K viewed from the cytoplasm. **d** Interactions of the cytoplasmic domain with the TM domain and the C-terminal helices (shown as a molecular surface coloured as in (**a**), viewed from the cytoplasm

hairpin and the associated loops provide important interactions between the cytoplasmic domain and the β9−β10 hairpin on the TM domain (Fig. 3d and Supplementary Fig. 5a). The other surface of the β-sheet is tightly packed against α8, part of an α-helical hairpin formed by α7 and α8, extensions of TM6 and TM7 (Supplementary Fig. 5b). This interaction links the position of the cytoplasmic domain to the TM helices. In the region between β3 and α3 the structures of hTMEM16K, hTMEM16F, mTMEM16A and nhTMEM16 differ significantly: hTMEM16K has a relatively short α3 and a β4−β5 hairpin that interacts directly with the C-terminal α10[A] and α10[B] helices (Supplementary Fig. 5c−e), linking the position of the cytoplasmic domain to the dimer interface helices.

The TM domain of TMEM16K is C-terminal to the cytoplasmic domain and has ten TM α helices with TM3 to TM7 forming the lipid scramblase 'groove' and TM6 to TM8 forming the binding site for two $Ca^{2+}$ ions, seen in other TMEM16s (Figs. 3a, b, 4a−e). In hTMEM16K, as in other TMEM16 scramblases, the groove is lined by a series of charged and hydrogen bonding sidechains that could interact with lipid headgroups (Fig. 4b).

The C-terminal region of TMEM16K forms the dimer interface, consisting of TM10 in the membrane region and its domain-swapped, cytoplasmic extension, α10 (Fig. 4c, d, Supplementary Fig. 6a−f). There is a large dimer interface cavity between TMs 3, 5, 9, 10 and TM10[B], which contains several monoacylglycerol (MAG)7.9 lipids in the LCP structure (Supplementary Fig. 6e). In afTMEM16 we proposed that this cavity could play a role in deformation of the membrane[37], it remains to be seen if it plays a similar role in TMEM16K.

**MD simulations on this TMEM16K conformation show scrambling.** We used MD simulations to investigate how lipids traverse the TMEM16K groove. Coarse-grained (CG) MD simulations allow us to sample the longer time scales necessary to investigate lipid scrambling events. Of note is that of the >3500

membrane proteins analysed for CGMD lipid scrambling, as part of the MemProtMD database[24,41], nhTMEM16 is one of the very few proteins to display scrambling activity[24]. CGMD simulations for hTMEM16K revealed similar lipid translocation events in both directions through the groove (Fig. 5a−e).

To expand on these observations, we converted a mid-flipping CG state of TMEM16K in a 1-palmitoyl-2-oleoyl-glycero-3-phosphocholine (POPC) membrane to an atomistic description. Here, we also see lipid transport along the groove (Fig. 5f−g Supplementary Fig. 7a), with the full translocation pathway sampled. Note that whilst the 2.1 μs of atomistic simulation is sufficient to observe lipids entering, moving along, and exiting the groove, no individual lipid fully samples the entire pathway. We estimate from the data that the atomistic rate is ca. 10-fold slower than the CG, in line with previous observations about kinetics in Martini[42].

As the scramblase activity was shown to be 20-fold faster when TMEM16K was placed in liposomes containing short chain lipids, we ran MD simulation to investigate the effect of changing the lipids from POPC and 1-palmitoyl-2-oleoyl-glycero-3-phosphoglycerol (POPG) to 1,2-dimyristoyl-sn-glycero-3-phosphoglycerol (DMPC)/1,2-dimyristoyl-sn-glycero-3-phosphoglycerol (DMPG), while maintaining the open groove conformation of TMEM16K in silico. The apparent rate of scrambling increased significantly (Supplementary Fig. 7b) in both directions with the shorter chain lipids, a result that is consistent with the lipid scramblase assay results.

Atomistic analysis identified a large number of residues along the groove facing sides of TM4 (Tyr360, Ser363, Ala367, Glu371 and Arg375), TM5 (Cys412, Ser415 and Ile419), TM6 (Leu428, Gln431, Ser432, Leu436, Ser440 and Asn444) and TM7 (Leu503 and Tyr 507) (Fig. 4b) that have a substantial sidechain to lipid head group interaction occupancy (>30% of simulation time). These scrambling residues generally retain the hydrophilic profile of those seen in other TMEM16 scramblases. However, there are very few highly conserved residues in the groove across the TMEM16 family, perhaps reflecting the general lack of strict lipid

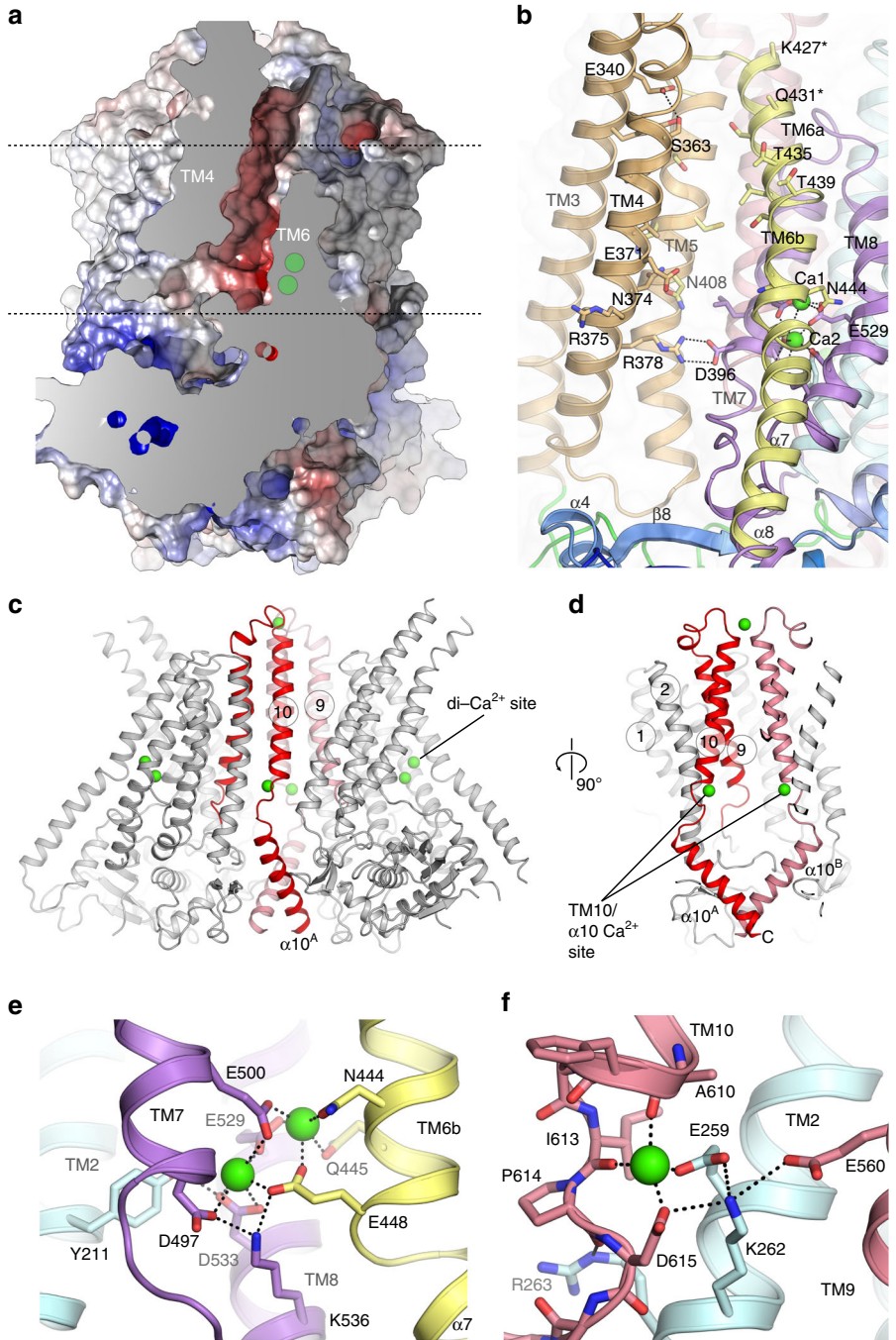

**Fig. 4** Structure of the TM domain region, scramblase groove and $Ca^{2+}$ binding sites. **a** Sliced molecular surface showing TMEM16K's scramblase groove. Approximate positions of membrane and two $Ca^{2+}$ ion site indicated by dotted lines and green circles. **b** Ribbon representation of the groove (coloured as per Fig. 3b). Selected residues that were in contact with the lipid headgroups in MD simulations are shown as sticks and labelled. **c** Location of the $Ca^{2+}$-binding sites (green) in TMEM16K, and the dimer interface formed by TM10 and α10 contains the third $Ca^{2+}$ ion, in a second $Ca^{2+}$-binding site. **c, d** Ribbon representation of dimer highlighting positions of TM10-α10 helices (red). Zoomed in view of **e** the TM6-8 two $Ca^{2+}$-binding site and of **f** the TM10/ α10 $Ca^{2+}$-binding site

specificity and differences in the roles of the different family members in various cellular environments. Ser363 is of particular interest, as in the known scramblases (afTMEM16, nhTMEM16, TMEM16K and TMEM16F) it is either Ser or Thr, whereas in TMEM16A it is a Val (Supplementary Fig. 2). Interestingly, swapping from Val to either Ser or Thr or vice versa switches TMEM16A to a scramblase and nhTMEM16 to a channel, suggesting that this residue could be important for scrambling in TMEM16K[9].

The relative lipid distributions in the MD simulations (as indicated by phosphate and choline occupancy) show only minimal variance along the length of the scramblase groove over the time course of the simulation (Fig. 5e, f). This is in contrast to previously published simulations performed on nhTMEM16, which identify a more punctuated series of local maxima for phosphate occupancy along the groove[9,23]. Taken together the MD data clearly indicate that the TMEM16K groove serves as a lipid translocation pathway.

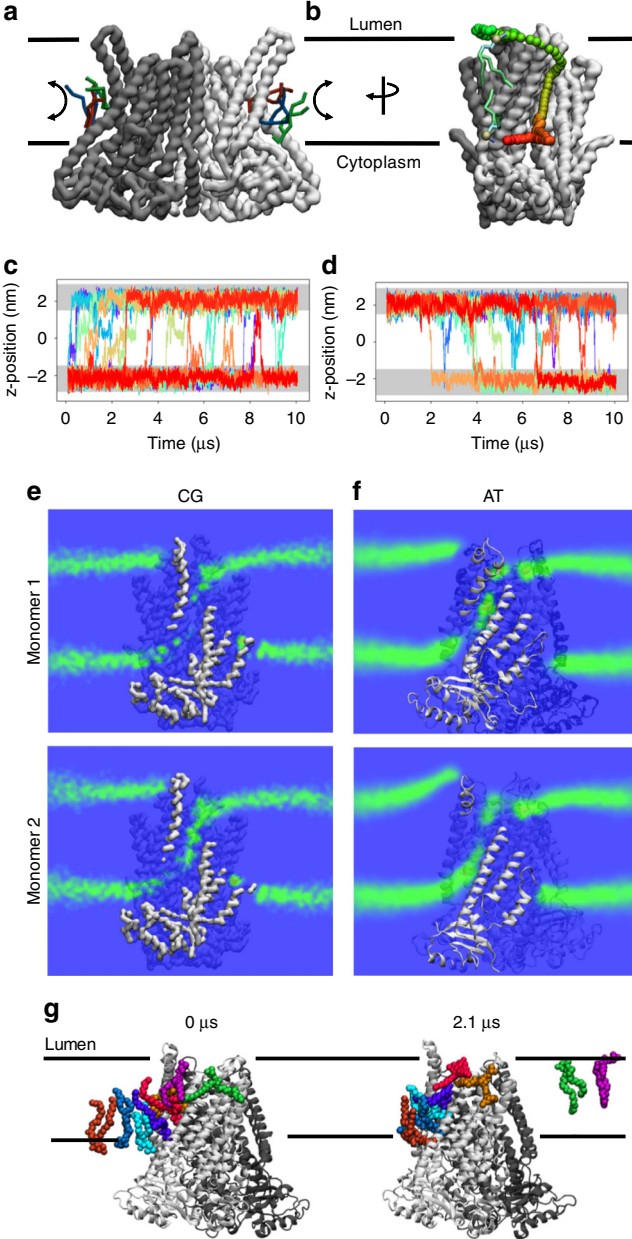

**Fig. 5** CG and atomistic MD simulations of TMEM16K structures reveal lipid scrambling. **a** View of CG TMEM16K after ~8 μs simulation. The protein backbone surface (grey and white) and scrambling lipid molecules (blue, red and green sticks) are shown, **b** as panel (**a**), but rotated by 90° and showing the pathway of a single lipid. The before and after poses are shown as green, blue and cyan sticks, with the phosphate head group particle shown as a gold/grey sphere. The scrambling pathway is denoted by red-to-green coloured spheres from cytoplasmic to luminal side. **c**, **d** Traces showing quantification of CG lipid scrambling, across the z-axis, with the position of phosphate beads in each membrane leaflet shown in light grey. The traces show **c** lipids that flip from cytoplasmic to lumenal leaflets, and those that flip in the opposite direction, **d**, **e** Density of lipid headgroups along a plane through the membrane, as computed over 10 μs of CG simulation. Density is scaled from blue to green, with a scrambling pathway clearly visible along the TMEM16K hydrophobic groove. **f** As panel (**e**) but of 2.1 μs of atomistic simulation, and calculating the densities of all non-hydrogen head group atoms. The densities reveal a similar pattern to the CG data. **g** Lipids engaged with the TMEM16K translocation pathway in a 2.1 μs atomistic simulation, coloured as per panel (**a**)

**TMEM16K has a two Ca²⁺-site and a TM10-α10 Ca²⁺- site.** Regulation of TMEM16 proteins by cytosolic Ca²⁺ is thought to involve the conserved two Ca²⁺ binding site situated between TM6, TM7 and TM8. This site is preserved in hTMEM16K (Fig. 4a–e), with anomalous difference maps confirming the presence of two Ca²⁺ ions (Supplementary Fig. 8,a, b) and is formed from the sidechains of a series of highly conserved residues (Fig. 4e, Supplementary Fig. 2). Mutation of these residues reduced Ca²⁺ binding and stability in mTMEM16K[43].

Surprisingly, our anomalous difference electron density maps revealed an additional Ca²⁺ ion bound near the dimer interface between TM2, TM10 and α10, where TM10 emerges from the membrane (Fig. 4c, d, f, Supplementary Fig. 8,a, c). A fourth Ca²⁺ ion site on the ER side of the membrane at the dimer interface (Supplementary Fig. 8d) was seen only in the LCP crystallisation, not in the VD or other structures, so this site is likely to be the result of the crystallisation conditions, not a characteristic of the protein or this conformation. The TM10 and α10 Ca²⁺ ion interacts with the sidechains of Glu259 and Asp615 and the backbone carbonyls of Ala610 and Ile613, terminating the TM10 helix (Fig. 4f). In one case of ataxia associated with compound heterozygous TMEM16K mutations, one TMEM16K allele has an Asp615Asn mutation[13], which could partially impair Ca²⁺-binding at the TM10-α10 site (Fig. 4f). However, initial experiments suggested that the Asp615Asn mutant did not alter lipid scrambling compared to WT TMEM16K in the presence of either saturating or 0 mM Ca²⁺ (Supplementary Fig. 3e, f). Therefore, further work is required to characterise the effect of this mutation. Interestingly, the residues coordinating this site are conserved in mammalian, but not fungal TMEM16s (Supplementary Fig. 2). Indeed, inspection of the mTMEM16A electron density maps showed density at this location consistent with a Ca²⁺ ion (Supplementary Fig. 8j), while a recent publication confirmed the presence of a Ca²⁺ ion at this site in TMEM16F[30].

**TMEM16K cryo-EM structures reveal a closed groove state.** To understand whether TMEM16K can adopt alternative conformations and how changes in Ca²⁺ concentration affect the structure, we obtained three electron cryomicroscopy (cryo-EM) structures for TMEM16K in the presence of (i) 2 mM ('high') Ca²⁺, (ii) 430 nM ('low') Ca²⁺ and (iii) in the absence of Ca²⁺ (in 10 mM ethylene glycol-bis(β-aminoethyl ether)-N,N,N′,N′-tetra-acetic acid (EGTA), 'Ca²⁺-free'). These data sets had resolutions of 3.5, 4.2 and 5.1 Å respectively (Supplementary Figs. 9–11, Supplementary Table 3). The 'low' Ca²⁺ concentration sample was obtained without addition of either Ca²⁺ or EGTA during purification. As the concentration of Ca²⁺ in our buffers was found to be 431.5 ± 73.5 nM: (n = 3, s.e.m. estimate of error), we refer to this as a 430 nM Ca²⁺ sample, although the protein itself has additional bound Ca²⁺(see Methods section for further details). All the protein samples used in cryo-EM were purified in the detergent undecyl maltoside (UDM) with cholesteryl hemi-succinate (CHS) added to improve protein stability: the same lipid detergent combination that was used for the VD crystallisation. The low and high Ca²⁺ structures are remarkably similar (mainchain r.m.s.d. of 0.42 Å); therefore below we describe the higher resolution 2 mM Ca²⁺ structure, unless otherwise stated.

All three structures showed a dramatic change in conformation with extensive movements from the cytoplasmic to the ER side of the membrane, converting the classic scramblase open groove arrangement described above, to a state where the groove is closed at the ER end. In the groove region, this conformation resembles the Ca²⁺-bound state of the mTMEM16A channel[25–27], and the recently published TMEM16F[30], afTMEM16[37] and nhTMEM16[29] closed groove structures (Fig. 6f, h, Supplementary

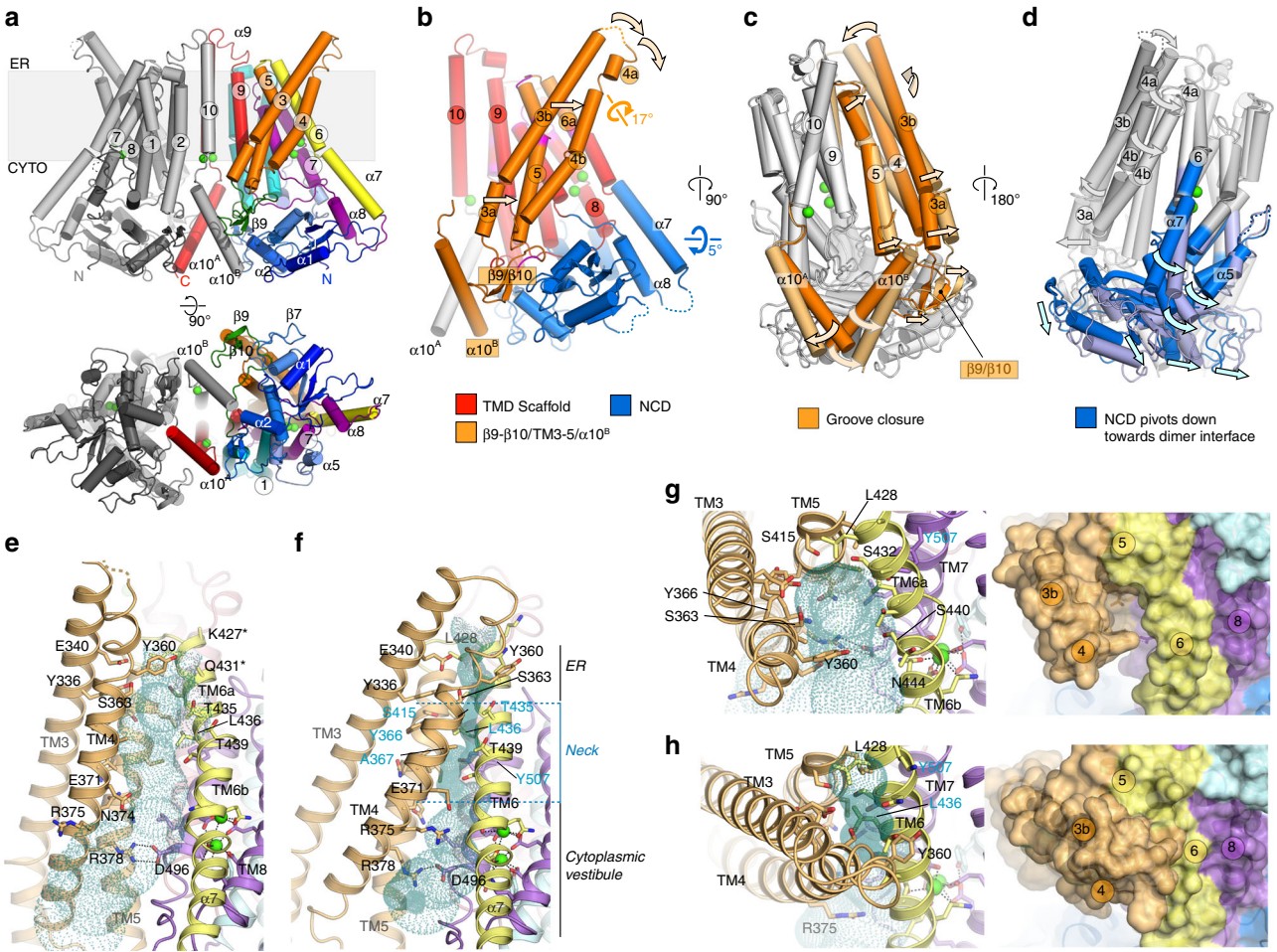

**Fig. 6** Cryo-EM structure reveals a closed scramblase conformation of TMEM16K. **a** Structure of TMEM16K in 430 nM Ca$^{2+}$ obtained by cryo-EM. Structural elements are coloured and viewed as in Fig. 2a. **b** Scaffold (red), TM4-6/α10$^B$ (orange) and NCD/α7-α8 regions (blue) mapped onto the monomer fold as derived from domain motion analysis. **c, d** Relative motions of **c** the TM4-6/α10$^B$ region (orange) and **d** the NCD/α7-α8 region, with the domains from the open groove crystal (dark colours) and closed groove 430 nM Ca$^{2+}$ cryo-EM (lighter colours) structures viewed looking onto the **c** dimer interface and **d** scramblase groove. **e−h** Schematic representation of the scramblase groove in the crystal structure (**e, g**) and 430 nM Ca$^{2+}$ structure (**f, h**). The groove is viewed perpendicular to the membrane normal (**e, f**) and from the ER luminal face (**g, h**). Pale green dotted surface represents the groove/channel profile as calculated by HOLE[77]

Fig. 12a). Although the Ca$^{2+}$ concentration is reduced from 100 mM in the X-ray structures to either 2 mM or 430 nM for the cryo-EM structures, there is clear density for Ca$^{2+}$ ions at both the TM6-8 site and the TM10-α10 site (Supplementary Fig. 8e−h). Therefore, these conformational changes are not caused by loss of Ca$^{2+}$ binding at either site.

The change from an open to a closed scramblase groove state is associated with extensive movements of the cytoplasmic domain and several transmembrane helices, leading to the ER end of the TM3−TM4 α-helical hairpin blocking the ER end of the groove (Fig. 6, Supplementary Fig. 12). Within each TMEM16K monomer, there are two significant independent domain movements, one involving the region from β9 to TM5 and other involving the cytoplasmic domain[44]. There is a semi-rigid TM scaffold sub-domain consisting of TM1-2, TM7-8 and TM9-10, and the two Ca$^{2+}$ ions (Fig. 6b). The rest of the structure is divided into two parts that move separately. The first unit, referred to here as the TM3−5 unit, consists of the β9–β10 hairpin, TM3, TM4 and TM5, as well as the domain-swapped α10 from the other chain. This whole unit rotates by ~17° relative to the scaffold domain, bringing α10 and the β9–β10 hairpin towards the front of cytoplasmic domain, as shown by torsional analysis using DYNDOM[44] (Fig. 6b–d, Methods section and

Supplementary Fig. 12, Supplementary Movie 1). This allows TM3, TM4 and TM5 to move up, with TM3 and TM4 pivoting around TM5. This causes TM4 to move up to 10 Å at its ER end, so that it can fold over the groove, blocking the ER end of the groove (Fig. 6c, e−h). The second unit, consisting of the N-terminal cytoplasmic domain up to α6 and the α7−α8 helices (cytoplasmic extensions of TM6 and 7), rotates downwards by 5° relative to the TM scaffold structure (Supplementary Fig. 12f). This causes the cytoplasmic domain to move down and away from the TM helices and the TM3−5 unit.

TM6 lies between the scaffold and the TM3−5 unit, and is divided into TM6a, which moves ~1 Å with the TM3−5 unit, and TM6b, which forms part of the two Ca$^{2+}$-binding site and is associated with the TM scaffold domain (Supplementary Fig. 12e, f). TM6 therefore provides a flexible buffer between the scaffold and the TM3−5 unit, and has an ordered α7 extension which interacts with α8 and the N-terminal cytoplasmic domain (Fig. 6a, b, Supplementary Fig. 12e, f).

Unlike other TMEM16s, TMEM16K shows significant changes in the relative orientations of the two monomers that make up the dimer, when comparing the open and closed groove structures (Supplementary Fig. 12c). At the dimer interface, although the TM10 contacts are similar, the α10$^A$−α10$^B$ contact in the

cytoplasm is lost completely in the closed groove structure, such that the overall dimer interface area decreases by 30% (from 1760 $\text{Å}^2$ to 1236 $\text{Å}^2$ comparing open and closed groove structures) (Supplementary Fig. 6f–i). The relatively small changes seen when the monomers are compared are significantly larger when the TMEM16K X-ray and cryo-EM dimers are superimposed (Supplementary Fig. 12c). While the relative alignment of the TM10 dimer helices is the same in both structures, each of the other TM elements are offset by ~10° and the NCDs rotate by 10° compared to their positions in the open groove structure. These changes in domain orientation could contribute to a distortion of the membrane plane similar to that seen with afTMEM16 [37]. The MD simulations (Supplementary Fig. 7c, d) suggest that there could be some distortion of the planar membrane structure in the vicinity of the lipid entry route, although this is not as dramatic as was observed for afTMEM16 [37].

Interestingly, the TM3−5 unit forms one side of the hydrophobic dimer cavity. The presence of this cavity could provide space to accommodate movements of the TM3−5 unit, potentially even allowing greater opening of the groove, as recently proposed[35], to accommodate PEGylated lipids[36]. The dimer interface cavities in the TMEM16K structures are filled with detergent or lipid density in both open and closed groove states (Supplementary Figs. 6a−f, 9f−h). There is clear density for one complete lipid molecule in the 3.5 Å closed groove structure, lying adjacent to TM3, TM5 and TM9, in the cytoplasmic leaflet, a critical site for movement of the TM3−5 unit. It is conceivable that the nature of the lipids occupying the dimer cavity could affect the activity of TMEM16K, through direct interactions with TM helices, α10 and the NCD.

The closed scramblase groove has three distinct sections: a hydrophilic vestibule on the ER side of the membrane, a narrow neck and a cytoplasmic vestibule (Fig. 6f, h). The hydrophobic neck is lined with residues from TM4−7 (Tyr366, Ala367, Leu416, Ser415, Thr435, Leu436 and Tyr507), which together form a physical barrier to lipid movement (Fig. 6f, h). Unsurprisingly, we did not observe lipid scrambling in CGMD simulations of TMEM16K in this closed groove conformation (Supplementary Fig. 12j).

TMEM16F scramblase structures observed to date display a closed groove regardless of the presence $Ca^{2+}$ref. [30], raising the question of whether mammalian TMEM16 scramblases can undergo an open-closed rearrangement similar to that observed in the fungal afTMEM16 [37] and nhTMEM16 [29]. Our TMEM16K structures (Figs. 4, 6) suggest that this is the case, by providing evidence that in a mammalian TMEM16 scramblase the groove can undergo an open-closed conformational rearrangement. Moreover, in the case of TMEM16K, movements of the TM3−TM5 section are accompanied by extensive changes that extend from the cytoplasmic to the ER side of the membrane, changes that were not observed in other TMEM16 structures. These extensive rearrangements are shown not be dependent on changes in $Ca^{2+}$ binding.

**TMEM16K conformation in the absence of $Ca^{2+}$.** To investigate how $Ca^{2+}$ binding regulates the TMEM16K conformation we determined the cryo-EM structure of TMEM16K in 10 mM EGTA to 5.1 Å resolution (Fig. 7, Supplementary Fig. 11). This structure showed that the groove remains closed at the ER end as seen in both $Ca^{2+}$cryo-EM structures. However, the $Ca^{2+}$-free structure reveals additional rearrangements at the two $Ca^{2+}$-binding site (Fig. 7). The TM6-8 two $Ca^{2+}$-binding site expands, with TM6/α7 helices moving up and away from TM7/8 by 2−3 Å, and the C-terminal end of the α7 TM6 extension becoming disordered (Fig. 7a−d). These movements recapitulate

the changes seen in the other TMEM16 family $Ca^{2+}$-free structures[37] and most closely resembles the $Ca^{2+}$-free structures of nhTMEM16 and mTMEM16F in lipid nanodiscs (Fig. 7e). While the resolution of the $Ca^{2+}$-free structure is not sufficiently high to resolve whether calcium ions still occupy these sites, the movement of TM6, which provides three sidechain ligands for the dual $Ca^{2+}$ site, would suggest that they are not present. The two $Ca^{2+}$-binding site is integral to the scaffold domain and forms one side of the groove, so it is not surprising that conformational changes associated with loss of $Ca^{2+}$ affect scramblase activity. However, TMEM16K retains basal scramblase activity even in the absence of $Ca^{2+}$, suggesting that there is an additional, as yet unknown, open groove $Ca^{2+}$-free scramblase conformation (Fig. 8).

## Discussion

In this study, we aimed to investigate the structural and functional properties of TMEM16K, together with its subcellular localisation, to further our understanding of its role in healthy and diseased cells. Our structures of TMEM16K have allowed us to explore some of the conformational changes that occur when the lipid scramblase groove opens and closes (Fig. 8). These changes are propagated from the cytoplasmic to the ER face of the membrane through movements of two rigid bodies, one involving the $\alpha 10^B$, β9−β10 hairpin and the TM3−TM5 unit and the other involving the cytoplasmic domain and the α7−α8 helices. The first unit rotates and moves up so that TM3−TM4 blocks the groove, while the second unit moves down, providing space for the first unit to rotate. These changes in TMEM16K conformation are not caused by changes in $Ca^{2+}$ binding, as all three $Ca^{2+}$-binding sites are occupied in both the 2 mM and 430 nM $Ca^{2+}$ structures. Complete removal of $Ca^{2+}$ leads to further changes in the vicinity of the $Ca^{2+}$ binding sites and at the lipid translocation pathway. In the $Ca^{2+}$-free conformation the groove is closed, so lipids will not be translocated. Since TMEM16K scrambling is reduced, but not completely ablated, in the absence of $Ca^{2+}$ it is possible that there is an additional, as yet unobserved, $Ca^{2+}$-free conformation with an open groove (Fig. 8). We hypothesise that both the $Ca^{2+}$-bound and $Ca^{2+}$-free states exist as an equilibrium between open and closed groove conformations; when $Ca^{2+}$ is bound the open state is preferred, whereas when there is no $Ca^{2+}$ bound, the closed state is favoured. Further, our data suggest that one additional factor affecting the equilibrium between open and closed states is the lipid composition of the membrane. The structures show changes in the lipid and detergent densities in the dimer interface cavities, although higher resolution structures will be needed to define the details of the lipid interactions.

Our results show that TMEM16K resides mainly in the ER, and that it has dual activity as a $Ca^{2+}$-regulated scramblase and non-selective ion channel. We found that scramblase activity critically depends on the nature of the lipids in the membrane and membrane thickness. The scramblase activity is greatly augmented in the presence of shorter chain lipids, which form thinner bilayers, reminiscent of the ER membrane (Fig. 2c, d). Continuous scrambling and/or flip/floppase activities are needed in ER membranes to redistribute lipids that are synthesised on its cytoplasmic face. Our results suggest that the scramblase activity of TMEM16K, possibly together with TMEM16E[11,31] and other opportunistic GPCR scramblases[45−47], could contribute to the scrambling activity required to maintain the symmetrical lipid distribution of the ER membrane. In contrast, the PM's highly asymmetrical lipid distribution is broken down only under extreme conditions such as apoptosis so that PM scramblases, like TMEM16F, may require tight regulation[6,7,34]. We propose that

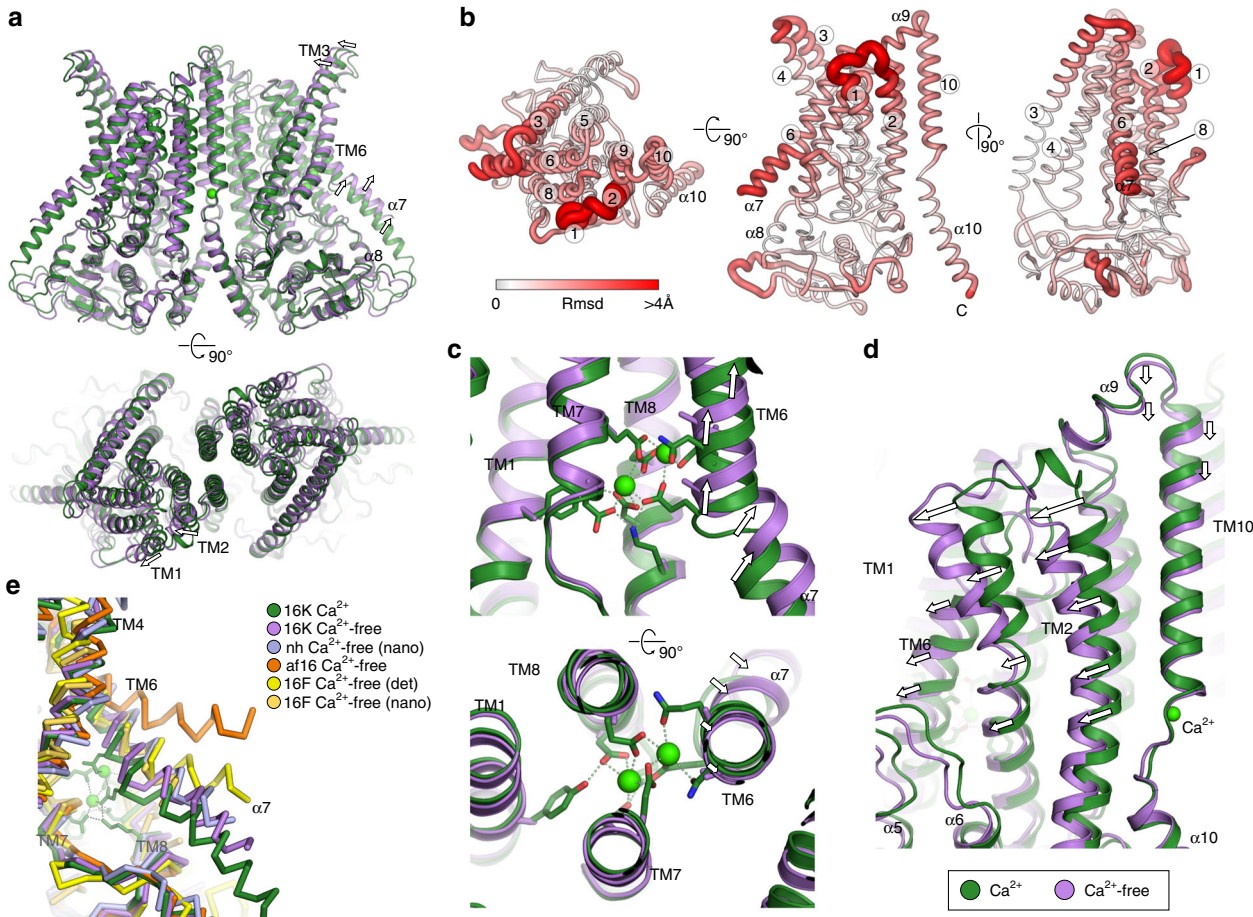

**Fig. 7** Comparison of the Ca²⁺-free and 2 mM cryo-EM structures of TMEM16K. **a** Overall structure of Ca²⁺-free state (purple), superimposed on the Ca²⁺ structure (green). **b** Global conformational differences between Ca²⁺ and Ca²⁺-free structures. Schematic representations of the r.m.s. deviation (rmsd) in mainchain atomic positions mapped onto the monomer Ca²⁺-free structure. Monomers were superposed using all atoms with LSQKAB (CCP4). The monomer is viewed from the luminal face (left), perpendicular to the membrane normal (middle) and scramblase groove face (right). The thickness and colour of the tube reflects the magnitude of the r.m.s.d. between the two structures. The main differences are localised in TM1/2, TM6, α7, TM9 and TM10. **c** View of the two Ca²⁺-binding site, showing overall movement of TM6. **d** View of the TM10-α10 Ca²⁺-binding site, highlighting the lateral movement of TM1-2, TM6 away from the dimer interface. **e** Comparison of the Ca²⁺-free structure with those from *nh*TMEM16 (PDB: 6qm4), afTMEM16 (PDB: 6dz7), and mTMEM16F in detergent (PDB: 6qpb) and nanodiscs (PDB: 6qpi). The 2 mM TMEM16K Ca²⁺ structure with an intact dual calcium binding site (green) is shown for reference

the dependence of TMEM16K on membrane thickness might provide a failsafe mechanism to prevent this protein from collapsing the membrane asymmetry should it be localised to non-ER membranes. The structures of the human TMEM16K presented here, together with the conformational changes seen in other TMEM16 scramblases (nhTMEM16 and afTMEM16[37]), demonstrate that there is extensive flexibility in TMEM16 structures, allowing them to adopt a range of activation states, potentially triggered by diverse stimuli such as membrane thickness, lipid composition, posttranslational modification or cofactor binding.

TMEM16K has been associated with several cellular phenomena, including spindle formation[20], Ca²⁺ signalling[21], volume regulation[10] and apoptosis[20,21]. Our work establishes its role as an ER scramblase, suggesting that the relationship between these cellular functions and lipid distributions warrants investigation. These results provide a structural framework to understand how compound heterozygous missense mutations found in patients with SCAR10 ataxia could affect TMEM16K function (Fig. 9). One mutation, Asp615Asn[13], lies in the TM10-α10 Ca²⁺-binding site (Fig. 4f), where it forms one of the Ca²⁺ ion ligands and an ionic bond with Lys262, linking TM2 to the connection between

TM10 and α10. However, activity assays on this mutant protein did not show changes in scramblase activity, highlighting the need to assess the function of mutants as well as wild-type (WT) proteins to understand the effects of mutations. There are also four missense mutations that introduce residues with very different sized sidechains into key positions between α helices, likely disrupting helix packing (Gly229Trp[14], Leu510Arg[12]) (Fig. 9b, c) or interfering with conformational changes (Phe171Ser[14], Phe337Val[14]) (Fig. 9d, e). The observation that TMEM16K truncations and missense variants lead to SCAR10 and our discovery that TMEM16K is an ER scramblase, suggest that the underlying cause of this ataxia could be associated with incorrect lipid distributions in ER and other membranes. This hypothesis provides a new direction for research into the underlying biology of these ataxias and the development of novel therapies.

## Methods

**Cell culture for electrophysiology and confocal imaging**. *TMEM16A* (Genbank NM_178642), *TMEM16F* (NM_175344) and *TMEM16K* (NM_018075) inserted into the pcDNA3.1 vector were used in this study. Human embryonic kidney 293T cells (HEK-293T, ATCC collection: ATCC CRL 1573) were grown in DMEM-F12 medium (D8437, Sigma) supplemented with 10% fetal calf serum and 0.05 mg/100 ml gentamicin. HEK-293T cells were transfected with 0.6 µg of

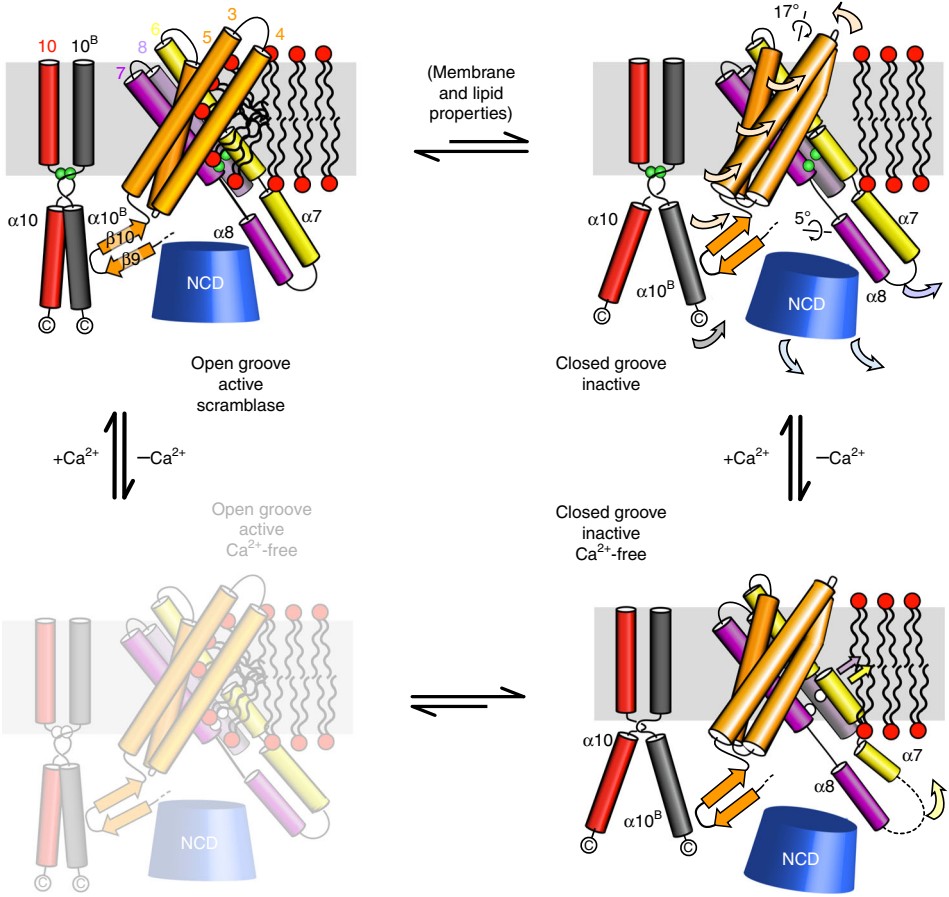

**Fig. 8** Schematic of conformational changes in TMEM16K. Schematic summary of TMEM16K conformational states, with the N-terminal cytoplasmic domain (NCD) shown as a blue cone and lipids with headgroups (red) and aliphatic chains (black). Conformations for which the structure is unknown are shown with higher transparency, and are included for completeness of the schema

TMEM16A, TMEM16F or TMEM16K and 0.2 μg of CD8 construct using Fugene HD (Promega, UK) according to the manufacturer's instructions. Cells were used ~12–24 h after transfection. Transfected cells were visualised using the anti-CD8 antibody-coated beads method[48]. Mock-transfected cells refer to cells treated with transfection reagent only. COS-7 (ATCC collection CRL-1651) and U2OS (U2OS FlpIn/T-Rex cells were a kind gift of M. Gyrd-Hansen, Ludwig Institute, Oxford[49,50]) cells used for immunostaining were maintained in DMEM high glucose media (D6546, Sigma) supplemented with 10% fetal calf serum and 0.05 mg/100 ml gentamicin and grown at 37 °C in 5% $CO_2$. For each cell line, solutions were changed every 3–4 days and cells were split to 1/10 when they reached 70–80% confluence.

**Electrophysiology**. TMEM16s currents were measured with the whole-cell configuration of the patch-clamp technique using an Axon 200B amplifier (Molecular Devices, USA) controlled with GE-pulse software (http://users.ge.ibf.cnr.it/pusch/programs-mik.htm). Currents were low-pass filtered at 2 kHz and sampled at 10 kHz. Pipettes were prepared from borosilicate glass capillary tubes (Harvard Apparatus, USA) using a PC-10 pipette puller (Narishige, Japan). Pipette tip diameter yielded a resistance of ~2–3 MΩ in the working solutions. The bath was grounded through a 3 M KCl agar bridge connected to a Ag-AgCl reference electrode. Experiments were conducted at 20–22 °C. The cell capacitance was assessed by measuring the area beneath a capacitive transient elicited by a 10 mV step or via the cell capacity compensation circuit of the amplifier. Current density was obtained by dividing the current amplitude for the cell capacitance.

The extracellular solution contained: 150 mM NaCl, 1 mM $CaCl_2$, 1 mM $MgCl_2$, 10 mM glucose, 10 mM D-mannitol and 10 mM HEPES; pH was adjusted to 7.4 with NaOH. The intracellular solution contained: 130 mM CsCl, 10 mM EGTA, 1 mM $MgCl_2$, 10 mM HEPES and 8 mM $CaCl_2$ to obtain $[Ca^{2+}]_i$ of ~300 nM; pH was adjusted to 7.3 with NaOH. The intracellular solution containing ~78 μM $[Ca^{2+}]_i$ was obtained by replacing EGTA with equimolar H-EDTA and by adding a total of 9 mM $CaCl_2$[51].

**Immunostaining**. Immunofluorescence staining was carried out using the (i) COS-7, (ii) U2OS, and (iii) HEK-293T cell lines. The HEK-293T and COS-7 cells were obtained from the ATCC collection (ATCC CRL 1573 and CRL-1651 respectively)

and the U2OS FlpIn/T-Rex cells were a kind gift of M. Gyrd-Hansen, Ludwig Institute, Oxford[49,50]. Briefly, cells directly plated and grown on glass coverslips were washed three times with cold phosphate-buffered saline (PBS) (Gibco, UK) and fixed in 4% paraformaldehyde in PBS for 20 min. After fixation, cells were washed twice with cold PBS. For staining of the ER, cells were permeabilised with 0.2% (v/v) Triton-X-100 (Merck, UK) in PBS for 5 min. Following this, cells were washed with cold PBS and then blocked with 0.2% fish skin gelatin (FSG) in PBS for 30 min at room temperature. Coverslips were then incubated in PBS containing 0.2% FSG with the following antibodies: anti-FLAG (Sigma, clone M2, F1804, mouse monoclonal, 1:400) and anti-calnexin (CNX, Enzo Life Sciences, Adi_Spa-860, rabbit polyclonal, 1:400) or anti-Hrd1 (Abcam, EP7459, rabbit polyclonal, 1:400) for 1 h at room temperature. For U2OS cells, the following primary antibodies were used: anti-TMEM16K (Sigma, HPA051569, rabbit polyclonal, 1:100) or anti-KDEL (Enzo Life Sciences, 10C3, ADI-SPA-827, mouse monoclonal, 1:200). Coverslips were then washed three times with PBS and stained with appropriate secondary antibodies: anti-mouse IgG-Alexa 546 (Thermo-Fisher, 1:1000) and anti-rabbit IgG-Alexa 488 (Thermo-Fisher, 1:400) for 1 h at room temperature. Nuclei were stained with 4′,6-diamidino-2-phenylindole (DAPI, Sigma, D9542) before being mounted onto microscope slides using a drop of Vectashield mounting medium (Vector Laboratories, USA) and sealed with generic nail varnish. The microscope slides were stored in the dark at 4 °C until imaged.

Imaging analysis was performed by the LSM510 META confocal laser scanning system (Zeiss, UK) connected to an inverted AxioVert 200 microscope with a ×63 objective (Zeiss, UK), controlled by Zen 2012 (Blue edition) software (Zeiss, UK).

Co-localisation of TMEM16K with ER-resident proteins was quantitatively assessed by determining the Manders' overlap coefficient (MOC) and Pearson's correlation coefficient (PCC) using Fiji software with the Coloc2 plugin[52,53]. Two different MOCs were obtained (M1 and M2), which quantify the independent contributions of two selected channels to the pixels of interest. M1 represents the fraction of TMEM16K in regions containing the ER marker while M2 conveys the fraction of ER marker in regions of TMEM16K. PCCs were also calculated as a measure of pixel-by-pixel cross-correlation coefficients. MOC and PCC values range from 0 (no correlation) to 1 (complete overlap).

**Cloning and expression for structural and functional studies**. The *Homo sapiens TMEM16K* gene, which encodes the TMEM16K/anoctamin-10 protein, was

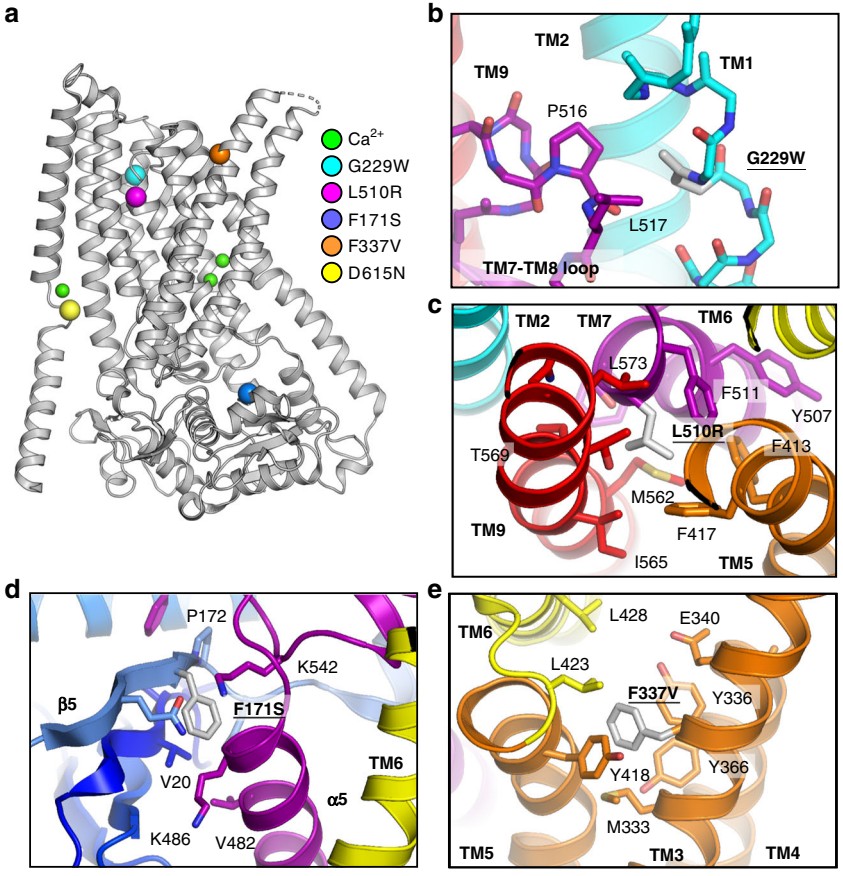

**Fig. 9** Missense variants in TMEM16K found in patients with SCAR10 ataxia. **a** Overview of TMEM16K with disease-associated missense mutation locations shown as coloured spheres and $Ca^{2+}$ ions shown as smaller green spheres. Local environment for the disease associated mutations: **b** Gly229Trp, **c** Leu510Arg, **d** Phe171Ser and **e** Phe337Val

provided by the DNASU Plasmid collection (https://dnasu.org/DNASU/Home.do). Coding DNA for the full-length human TMEM16K sequence (NM_018075), Met1 to Thr660 (Uniprot ID: Q9NW15), was cloned into the baculovirus transfer vector pFB-CT10HF-LIC (available from The Addgene Nonprofit Plasmid Repository) for expression in *Spodoptera frugiperda* (Sf9) cells (Thermo-Fisher Scientific, Cat. No. 11496015) using the primers shown in Supplementary Table 4. The vector adds a C-terminal TEV-cleavable His10-FLAG tag for purification. For mammalian expression, the same construct was also cloned into the pHTBV1.1-LIC baculovirus transfer vector (The BacMam vector backbone (pHTBV1.1) was kindly provided by Professor Frederick Boyce (Massachusetts General Hospital, Cambridge, MA) and adapted for ligation independent cloning in-house for expression in Expi239F cells (Thermo-Fisher Scientific, Cat. No. A14527) which similarly confers a TEV cleavable C-terminal His10-FLAG tag. In both expression cases, baculoviral DNA, produced by transformation of DH10Bac with either the TMEM16K-pFB-CT10HF-LIC or TMEM16K-pHTBV1.1-LIC transfer vectors, were used to transfect Sf9 cells to produce baculovirus particles for transduction. Virus was amplified by transducing mid-log Sf9 cells ($2 \times 10^6$ cells ml$^{-1}$) grown in Sf900II$^{TM}$ media with 2% fetal bovine serum. Cells were incubated on an orbital shaker for 65 h at 27 °C in 1 l shaker flasks. Baculovirus was harvested by centrifugation at $900 \times g$ for 20 min and the virus containing supernatant was used to infect 1 l of mid-log phase ($2 \times 10^6$ cells ml$^{-1}$) cultures of Sf9 cells in Sf−900$^{TM}$ II Serum Free Medium (Gibco/Thermo-Fisher) in a 3 l flask, which were then grown for 72 h at 27 °C on an orbital shaker. Cells were harvested by centrifugation at $900 \times g$ for 15 min, washed with phosphate-buffered saline (PBS), and pelleted again prior to flash freezing in liquid $N_2$, then stored at −80 °C until needed.

For mammalian (Expi293F) expression, baculovirus were prepared in Sf9 cells as described for insect cell expression. 1 l of Expi293F cell cultures ($2 \times 10^6$ cells ml$^{-1}$) in Freestyle 293$^{TM}$ Expression Medium (Thermo-Fisher) were infected with high-titre P3 baculovirus (3% v/v) in the presence of 5 mM sodium butyrate in a 2 l roller bottle (Biofil). Cells were grown in a humidity-controlled orbital shaker for 48 h at 37 °C with 8% $CO_2$ before being harvested by the same process as for insect cells.

**TMEM16K purification**. Sf9 cell pellets containing heterologously overexpressed protein destined for crystallographic analysis were resuspended in 30 ml l$^{-1}$ equiv. original cell culture Buffer A (20 mM HEPES pH 7.5, 200 mM NaCl, 5 % glycerol

v/v, 2 mM tris(2-carboxyethyl)phosphine (TCEP)) and lysed by two passes through a Emulsiflex C5 homogeniser (Avestin). A 10:1 mixture of *n*-Undecyl-β-D-Mal-topyranoside (UDM)/cholesteryl hemisuccinate (CHS) was added to the lysate giving a final concentration of 1% (w/v) for UDM and 0.1% for CHS, and incubated at 4 °C for 1 h on a roller. Insoluble material was removed by centrifugation at $32,000 \times g$ for 1 h at 4 °C. The supernatant was supplemented with imidazole pH 7.5 to a final concentration of 5 mM, and then a 50% slurry (v/v) pre-equilibrated Talon$^{TM}$ resin (1 ml slurry per l original culture volume) was added. The suspension was then incubated at 4 °C on a roller for 1 h. Talon resin was collected by centrifugation at $900 \times g$ for 15 min and transferred to a gravity column where the remaining liquid was allowed to flow through. Subsequently, all buffer solutions were supplemented with 0.045%:0.0045% (w/v) UDM/CHS. The resin was washed with 25 column volumes of Buffer A with 20 mM imidazole, pH 7.5. TMEM16K was eluted with Buffer A + 250 mM imidazole. Peak elutions were then exchanged into Buffer A using Sephadex PD-10 desalting columns (GE Life Sciences). The protein's C-terminal 10xHistidine-FLAG tag was removed by overnight incubation with 5:1 (w/w) TMEM16K:TEV protease. TEV protease and contaminants were removed by adding 1.5 ml pre-equilibrated 50% Talon resin and batch binding at 4 °C for 1 h on a rotator. The resin was removed by passing over a gravity column, with the resulting flow-through containing cleaved TMEM16K. Cleaved TMEM16K was concentrated to <1 ml using a Vivaspin 20 centrifugal concentrator (GE Life Sciences) and further purified by size exclusion chromatography on a Superose 6 size exclusion column in Buffer A.

TMEM16K purification for cryo-EM data collection was similar to that used for crystallisation, except that Buffer A was substituted for Buffer B (20 mM HEPES pH 7.5, 150 mM NaCl, 2 mM TCEP), with lower salt and with the glycerol removed, to improve the background on cryo-EM images. Purification from transduced Expi293F cells expressing wild-type or mutant TMEM16K was identical to the protocol used for purification from Sf9 cells.

The concentration of $Ca^{2+}$ in buffers and protein-containing solutions was determined using Calcium Green$^{TM}$-1 fluorescent indicator (Thermo-Fisher Scientific) according to the manufacturer's instructions. $Ca^{2+}$ concentration was calculated from a standard curve derived from solutions of known $Ca^{2+}$ concentration prepared using the Calcium Calibration Buffer Kit #1 (Thermo-Fisher Scientific) according to the manufacturer's instructions. The sample described as 430 nM was purified without the addition of either $Ca^{2+}$ or EGTA, so

the $Ca^{2+}$ present in the sample was obtained from the cells and the purification buffers. We tested the concentration of $Ca^{2+}$ in our size exclusion chromatography buffers and found that it was $430 \pm 73.5$ nM ($n = 3$, s.e.m.), so we refer to this sample as being in 430 nM $Ca^{2+}$, although it does of course have additional $Ca^{2+}$ associated with the protein. We measured the concentration of $Ca^{2+}$ after removing the protein from a solution containing 1 mgml$^{-1}$ protein by using a concentrator with a molecular weight cut off of 10 kDa. The eluate contained $1020 \pm 253$ nM $Ca^{2+}$ ($n = 3$, s.e.m.). This slightly higher concentration could be due to a portion of the protein denaturing during the separation process. We also tested the concentration of $Ca^{2+}$ directly in protein samples with and without denaturing the protein with heat. When the protein was denatured the $Ca^{2+}$ concentration was $37.2 \pm 10.7$ $\mu$M ($n = 6$, s.e.m.). The protein concentration is 13 $\mu$M, so with three $Ca^{2+}$ ions per monomer, we would expect 39 $\mu$M $Ca^{2+}$, which is similar to the concentration measured. Interestingly, when TMEM16K was not denatured, we measured a lower concentration of $Ca^{2+}$ in the solution ($2191 \pm 243$ nM, $n = 3$), suggesting that TMEM16K can compete with the dye for $Ca^{2+}$ ions. Since the dye has a $K_d$ of 190 nM for $Ca^{2+}$, it is possible that TMEM16K has an affinity for the dye that is higher than 200 nM. However, to accurately estimate the affinity for $Ca^{2+}$ requires further work.

**Western blots with the TMEM16K antibody**. Samples of both purified hTMEM16K and lysates from baculovirus-transduced HEK-293 (Expi239F) cells were run on SDS-PAGE and transferred to a polyvinylidene difluoride membrane using a Bio-Rad Criterion$^{TM}$ Blotter apparatus, according to the manufacturer's instructions. The membrane was briefly washed in TBS-T Buffer (20 mM Tris pH 8.0, 150 mM NaCl, 0.1% (v/v) Tween-20) before being blocked by incubation in TBST + 10 % (w/v) skim milk powder. The membrane was washed briefly in TBS-T before being incubated overnight at 4 °C with polyclonal Anti-ANO10 antibody (Sigma-Aldrich, Cat No: HPA051569), diluted 3000:1 in TBS-T. The antibody solution was decanted from the membrane, which was then washed 3 × 15 min in TBS-T. The membrane was then incubated with a secondary horseradish peroxidase-conjugated anti-rabbit IgG (Sigma-Aldrich, Cat No: A6667), diluted 1:2000 in TBS-T. The secondary antibody was decanted and the membrane washed 3 × 15 min in TBS-T. Blots were developed using the ECL system (Thermo-Fisher) according to the manufacturer's instructions and digitally imaged (Supplementary Fig. 1d).

**afTMEM16 expression and purification**. afTMEM16 was expressed and purified as described[33]. *S. cerevisiae* carrying pDDGFP2 [54] with afTMEM16 were grown in yeast synthetic drop-out medium supplemented with Uracil (CSM-URA; MP Biomedicals) and expression was induced with 2% (w/v) galactose at 30° for 22 h. Cells were collected, snap frozen in liquid nitrogen, lysed by cryomilling (Retsch model MM400) in liquid nitrogen (3 × 3 min, 25 Hz), and resuspended in buffer A (150 mM KCl, 10% (w/v) glycerol, 50 mM Tris-HCl, pH8) supplemented with 1 mM EDTA, 5 $\mu$g ml$^{-1}$ leupeptin, 2 $\mu$g ml$^{-1}$ pepstatin, 100 $\mu$M phenylmethane sulphonylfluoride and protease inhibitor cocktail tablets (Roche). Protein was extracted using 1% (w/v) digitonin (EMD biosciences) at 4 °C for 2 h and the lysate was cleared by centrifugation at $40,000 \times g$ for 45 min. The supernatant was supplemented with 1 mM MgCl$_2$ and 10 mM Imidazole, loaded onto a column of Ni-NTA agarose resin (Qiagen), washed with buffer A + 30 mM Imidazole and 0.12% digitonin, and eluted with buffer A + 300 mM Imidazole and 0.12% digitonin. The elution was treated with Tobacco Etch Virus protease overnight to remove the His tag and then further purified on a Superdex 200 10/300 GL column equilibrated with buffer A supplemented with 0.12% digitonin (GE Lifesciences). The *af*TMEM16 protein peak was collected and concentrated using a 50 kDa molecular weight cut-off concentrator (Amicon Ultra, Millipore).

**Liposome reconstitution and lipid scrambling assay**. Liposomes were prepared as described[33] from a 7:3 mixture of POPC and POPG. Lipids in chloroform (Avanti), including 0.4% w/w tail labelled NBD-PE, were dried under N$_2$, washed with pentane and resuspended at 20 mg ml$^{-1}$ in buffer B (150 mM KCl, 50 mM HEPES pH 7.4) with 35 mM 3-[(3-cholamidopropyl)dimethylammonio]-1-propanesulfonate (CHAPS). TMEM16K or afTMEM16 was added at 5 $\mu$g proteinmg$^{-1}$ lipids and detergent was removed using five changes of 150 mg ml$^{-1}$ Bio-Beads SM-2 (Bio-Rad) with rotation at 4 °C. Calcium or EGTA were introduced using sonicate, freeze, and thaw cycles. Liposomes were extruded through a 400 nm membrane and 20 $\mu$l were added to a final volume of 2 ml of buffer B supplemented with 0.5 or 2 mM CaCl$_2$ or 2 mM EGTA. The fluorescence intensity of the NBD (excitation-470 nm emission-530 nm) was monitored over time with mixing in a PTI spectrophotometer and after 100 s sodium dithionite was added at a final concentration of 40 mM. Data were collected using the FelixGX 4.1.0 software at a sampling rate of 3 Hz.

**Quantification of scrambling activity**. Quantification of the scrambling rate constants of TMEM16K and afTMEM16 were determined[36]. The fluorescence time course was fit to the following equation:

$$F_{tot}(t) = f_0 \left( L_i^{PF}(0) + (1 - L_i^{PF}(0)) e^{-\gamma t} \right) \\ + \frac{(1-f_0)}{D(\alpha+\beta)} \left\{ \alpha(\lambda_2 + \gamma)(\lambda_1 + \alpha + \beta) e^{\lambda_1 t} + \lambda_1 \beta(\lambda_2 + \alpha + \beta + \gamma) e^{\lambda_2 t} \right\}, \quad (1)$$

where $F_{tot}(t)$ is the total fluorescence at time $t$, $L_i^{PF}$ is the fraction of NBD-labelled lipids in the inner leaflet of protein-free liposomes, $\gamma = \gamma'[D]$ where $\gamma'$ is the second-order rate constant of dithionite reduction, $[D]$ is the dithionite concentration, $f_0$ is the fraction of protein-free liposomes in the sample, $\alpha$ and $\beta$ are the forward and reverse scrambling rate constants and

$$\lambda_1 = -\frac{(\alpha + \beta + \gamma) - \sqrt{(\alpha+\beta+\gamma)^2 - 4\alpha\gamma}}{2}, \quad (2)$$

$$\lambda_2 = -\frac{(\alpha + \beta + \gamma) + \sqrt{(\alpha+\beta+\gamma)^2 - 4\alpha\gamma}}{2}. \quad (3)$$

The free parameters of the fit are $f_0$ and $\alpha$ while $L_i^{PF}$ and $\gamma$ are experimentally determined from experiments on protein-free liposomes. In protein-free vesicles a very slow fluorescence decay is visible likely reflecting a slow leakage of dithionite into the vesicles or the spontaneous flipping of the NBD-labelled lipids. A linear fit used to estimate the rate of this process was estimated to be $L = (5.4 \pm 1.6) \times 10^{-5}$ s$^{-1}$ ($n > 160$)[35]. For TMEM16K, the leak is >2 orders of magnitude smaller than the rate constant of protein-mediated scrambling and therefore was considered negligible. All conditions were tested side by side with a control preparation of afTMEM16 in standard conditions.

**Flux assay**. Cl$^-$ flux assay was conducted as described below[8]. Liposomes were equilibrated in external buffer with low KCl (1 mM KCl, 300 mM Na-glutamate, 50 mM HEPES, pH 7.4) by spinning through a Sephadex G50 column (Sigma-Aldrich) pre-equilibrated in external buffer. To complete the experiment, 0.2 ml of the flow through from the G50 column was added to 1.8 ml of external solution and the total Cl$^-$ content of the liposomes was measured using an Ag:AgCl electrode after disruption of the vesicle by addition of 40 $\mu$l of 1.5 M *n*-octyl-$\beta$-D-glucopyranoside (Anatrace). The fraction of liposomes containing at least one active afTMEM16 or TMEM16K ion channel, A, was quantified as follows:

$$A = 100 \times \left( 1 - \frac{\Delta Cl}{\Delta Cl_{PF}} \right), \quad (4)$$

where $\Delta Cl$ is the change in [Cl$^-$] recorded upon detergent addition in protein-containing vesicles and $\Delta Cl_{PF}$ is the Cl$^-$ content of protein-free liposomes prepared in the same lipid composition on the same day.

**Statistical analyses**. Unless otherwise specified, all values are the mean ± s.e.m. and results are expressed as mean ± s.e.m. of $n$ (number of experiments). Statistical significance was determined with two-tailed unpaired $t$ test or one-way ANOVA with Bonferroni's post hoc test, as appropriate. For all statistical tests, $P$ values <0.05 were considered significant. SPSS (version 22; SPSS Inc., Chicago, IL, USA) or Excel (2013 Edition, Microsoft, Redmond, WA, USA) were used for statistical analysis.

For scramblase assays, rate constant values reported are the average of 6–9 individual fluorescence traces from three independent reconstitutions of TMEM16K and afTMEM16. For flux experiments, data points of A (% active liposomes) were averaged and points outside the 99% confidence interval were removed. The reported values are the averages of the remaining data points. This was necessary due to the variability in the recovery efficiency of liposomes. The scrambling rate constants and the % active liposomes are reported as mean ± st.dev.

**Crystallisation and X-ray data collection**. Purified TMEM16K was concentrated to 10–30 mgml$^{-1}$ using a Vivaspin 20 centrifugal concentrator with a 100 kDa molecular weight cut-off. Protein concentration was determined from the A280 using a Nanodrop spectrophotometer. TMEM16K was crystallised using both sitting drop VD crystallisation and in meso in LCP. VD crystals were grown in 0.1 M HEPES pH 7.0, 0.1 M calcium acetate, 22 % (v/v) PEG400, 0.05 mM C12E9 at a protein concentration of 10 mgml$^{-1}$. For LCP crystallisation, 30 mgml$^{-1}$ TMEM16K was combined with 1-(7Z-hexadecenoyl)-rac-glycerol (monoacyl-glycerol 7.9, Avanti Lipids) in a 1:1.5 ratio to form a lipid cubic phase. A 50 nl bolus of LCP-reconstituted TMEM16K was dispensed onto a glass LCP plate (Marienfeld, Germany) and overlaid with 800 nl of crystallisation solution. TMEM16K crystals grew in an LCP in a mother liquor containing 0.1 M MES pH 6.0, 0.1 M NaCl, 0.1 M CaCl$_2$, 30 % (v/v) PEG300. For both LCP and VD crystallisation, initial crystals appeared after 1 week and grew to full size within 3–4 weeks. All X-ray diffraction data were collected on the I24 microfocus beamline at the Diamond Light Source (Didcot, UK) from single crystals using a fine phi slicing strategy. Intensities were processed and integrated using XDS[55] and scaled using AIMLESS[56].

**Structural model building and refinement**. The initial data set collected on the LCP-derived crystals was phased by molecular replacement using PHASER[57] with the nhTMEM16 structure (PDB: 4WIS) as an initial search model. Phase improvement was performed using phenix.mr.rosetta[58]. The final model was built using COOT[59] and refined using BUSTER[60] using all data to 3.2 Å with appropriate NCS restraints. The final LCP model was subsequently used as a starting model for molecular replacement to solve the structure of detergent-solubilised TMEM16K using an anisotropic data set collected from a crystal grown using

sitting-drop vapour diffusion methods. The anisotropic nature of the VD data set prevented it being solved using nhTMEM16 (the only high-resolution structure at the time) as a molecular replacement search model. Processing of the VD data set was similar to that for the LCP crystals, with additional anisotropy correction performed using STARANISO[61]. The model geometry of the VD data set was improved by using LSSR target restraints to the LCP structure during BUSTER refinement in addition to TLS and NCS restraints (Supplementary Table 2).

The final LCP model encompasses residues Ser14 to Gln639. Several loops were poorly order and not modelled; residues 57–67 and part of the α7−α8 loop (residues 472–474) were disordered in chain A. The loops connecting either α5 and α6 to TM3−TM4 were poorly defined in both chains of the dimer and have also not been modelled. The final model also includes three $Ca^{2+}$ ions per monomer along with an additional $Ca^{2+}$ at the N-terminal end of TM10 on the dimer axis. The presence of $Ca^{2+}$ ions at these sites was indicated by peaks in both anomalous difference and PHASER log-likelihood gradient (LLG) maps calculated using a 3.4 Å data set with high multiplicity collected at a wavelength of 1.65 Å. Two $Ca^{2+}$ ions in each dimer lie at the canonical two $Ca^{2+}$ ion binding site and a third lies at the junction of TM10 and α10. All of these ions have bonds that are less than 2.5 Å to sidechains, suggesting the ions are not hydrated. The fourth $Ca^{2+}$ ion identified in the anomalous difference maps lies on the dimer twofold axis, binding to the backbone of the ER loop between TM9 and 10. The >4 Å interaction distances suggest that this ion is hydrated and is likely to be the result of the high (100 mM) $[Ca^{2+}]$ used in the crystallisation conditions (Supplementary Fig. 8d). This fourth ion is only present in the LCP data set, not in the vapour diffusion structure, so the presence of this ion is not necessary for the open conformation to be formed. In addition, elongated density within the dimer interface were interpreted as MAG7.9 lipids.

**Cryo-EM grid preparation**. Three microlitres aliquots of TMEM16K protein purified in UDM/CHS, at a concentration of 5 mgml$^{-1}$, was either applied to grid directly (430 nM $Ca^{2+}$ samples) or supplemented with 2 mM $CaCl_2$ or 10 mM EGTA, then were applied to glow-discharged holey carbon grids (Quantifoil R 1.2/1.3 Cu 300 mesh). Grids were blotted at 80–100% humidity for 3–5 s at 5 °C and plunge-frozen in liquid ethane using a Vitrobot Mark IV (FEI). $Ca^{2+}$-free TMEM16K grids were prepared in an identical manner using protein solution supplemented with 10 mM EGTA.

**Cryo-EM data collection**. Grid optimisation and preliminary data set collection was performed using the Tecnai F30 'Polara' transmission electron microscope (Thermo-Fisher) at Oxford Particle Imaging Centre (OPIC), Division of Structural Biology, Oxford operated at 300 kV, at liquid nitrogen temperature and equipped with a K2 Summit direct electron detector (Gatan) mounted behind an energy filter (GIF Quantum LS, Gatan). Screening was also performed using a Talos Arctica operating at 200 kV equipped with a Falcon3 detector (Thermo-Fisher) at the Central Oxford Structural Microscopy and Imaging Centre (COSMIC) Oxford University. Cryo-EM data collection was carried out on a Titan Krios at COSMIC, operating at 300 kV. Data were acquired with EPU software on a K2 Summit detector (Gatan) mounted behind an energy filter (GIF Quantum LS, Gatan) and operated in zero-loss mode (0–20 eV). For the 430 nM $Ca^{2+}$ and EGTA data sets, movies (20 frames, 8 s total exposure) were recorded in electron counting mode at a sampling rate of 0.822 Å per pixel and dose rate of between 6.36 and 7.06 e$^-$ Å$^{-2}$ s$^{-1}$ (total dose 51–56.5 e$^-$ Å$^{-2}$) with underfocus in the range from 1.0 to 3.0 μm. Details of all data sets are given in Supplementary Table 3.

**Cryo-EM data processing and model building**. All initial processing was carried out in RELION 2.0 [62] and 3.0 [63]. Frames in each movie stack were aligned and dose-weighted with MotionCor2 [64]. CTF parameters were estimated using CTFFIND 4.0 [65]. Dose-weighted stacks were subjected to semi-automatic particle picking using Gautomatch (https://www.mrc-lmb.cam.ac.uk/kzhang/Gautomatch/). Particles were picked from a subset of micrographs and 2D classified to produce class averages. The resultant representative class averages were then used as templates in Gautomatch for autopicking the full image sets. Particles were subjected to multiple (5–6) rounds of reference-free 2D classification. An initial model was generated ab initio in RELION and used as a reference of 3D classification with no symmetry imposed. Particles belonging to the best/highest resolution class(es) were pooled and taken forward into a second round of 3D classification with C2 symmetry. Finally, particles from the highest resolution class were used for auto-refinement. The 2 mM and 430 nM data sets were further processed in RELION 3 [63] to take advantage of new particle polishing and CTF refinement routines. In both cases, two rounds of individual particle CTF refinement interspersed with a single step of Bayesian polishing and 3D auto-refinement produced the best maps (Supplementary Figs. 9, 10). Further static 3D classification where no image alignment was performed with multiple (10) classes yielded a slightly better resolved reconstruction for the 2 mM $CaCl_2$ data set. The processing schema for all the data sets are shown in Supplementary Figs. 9–11. Local resolution estimation for each final reconstruction was performed with RELION[66]. Post-processing was carried out in RELION using a mask extended by 12 pixels with an additional 12-pixel soft edge that excluded the detergent micelle surrounding the protein. Conformational homogeneity appears to be well maintained within each data set and at no stage during 3D classification could we detect a subset of particles displaying a

more open groove conformation. For the most part, TM3 and TM4 remain well resolved within and between different 3D classes indicating a lack of structural heterogeneity for the region that is responsible for defining the extent of the scramblase groove.

Model building was carried out manually using the 3.4 Å TMEM16K LCP crystal structure (PDB: 5OC9) as a template. Briefly, chain A of the crystal structure was roughly fitted into the 3.5 Å resolution 2 mM $Ca^{2+}$ post-processed cryo-EM map in UCSF Chimera[67]. Subsequent model building was carried out in COOT[59]. The cytoplasmic domain was rotated into density and then appropriate sub-regions were fit to the density as rigid bodies. TM helices were fitted as rigid units or segmented where appropriate. The C-terminal α10 helix was rotated manually into position and sidechains were fitted using preferred rotamers. The remodelled chain A was superposed onto chain B and globally fitted to the cryo-EM density to create the symmetric dimer. Refinement was carried out at 3.5 Å resolution against the post-processed RELION map (low-pass filtered to 3.47 Å and sharpened with a B-factor of −112 Å$^2$) using phenix.real_space_refine (PHENIX v1.14) using non-crystallographic symmetry (NCS) constraints, rotamer restraints along with secondary structure restraints. Missing loop regions, not modelled in the crystal structure, were added and the calcium coordination was initially defined using phenix.metal-coordination. Subsequently the calcium coordination at both sites was maintained using distance restraints derived from the LCP X-ray structure. The final model encompasses all residues from Ser13-Lys641 and three $Ca^{2+}$ ions per monomer (Supplementary Table 3). Some of the lipid/detergent-like density around the TM domain at the dimer interface was modelled by six UDM molecules per monomer and a single phosphatidylcholine lipid. The precise identity of the lipid bound at the dimer interface between TM3, 5 and TM10 (of the adjacent monomer) is unknown.

The structure of the 430 nM $Ca^{2+}$ complex is very similar to the 2 mM $Ca^{2+}$ structure and only required minor adjustments to the poorly resolved loop regions between α5−α6 and α7−α8. The density for the detergent and lipid molecules was present in the map but less convincing at this resolution and so these heterogroups were removed. The resultant model was refined against the 4.2 Å RELION post-processed map (sharpened with a B-factor of −176 Å$^2$) using phenix. real_space_refine. Model geometry was maintained using NCS constraints in combination with reference model restraints to the 2 mM $Ca^{2+}$ structure.

The 2 mM $Ca^{2+}$ structure also served as a template for the low-resolution $Ca^{2+}$-free structure. The 2 mM $Ca^{2+}$ model was initially docked into the map using Chimera and then refined using phenix.real_space_refine with global minimisation and morphing using default restraints/constraints and additional secondary structure restraints. A cryo-EM map, low-pass filtered to 5.1 Å resolution and sharpened with a B-factor of −150 Å$^2$, was used for all refinement and model building. The TM1-loop-TM2 region (residues 222–254) and the N-terminal end of TM10 required additional manual rebuilding/fitting in COOT. The cytoplasmic α7−α8 region was poorly resolved and truncated between residues 463–474. At this resolution, there was no obvious density for the amino acid sidechains and the vast majority were truncated to their cbeta atoms. Only the sidechains of prolines and a few large hydrophobic residues were retained. In addition, the $Ca^{2+}$ ions were removed from the model. The resultant model was further refined with phenix. real_space_refine using NCS constraints, secondary structure restraints and reference model restraints to the 3.5 Å 2 mM $Ca^{2+}$ structure (Supplementary Table 3). All structures/refinement protocols were validated by randomising the final models by applying coordinate shifts of up to 0.3 Å using the noise function in PDBSET (CCP4). The resultant shifted models were then refined against the corresponding post-processed half1 maps. Model-to-map Fourier shell correlations (FSCs) were then calculated with phenix.mtriage using either the final refined model against the full map (FSCsum) or the randomised half map1 refined model against either the half1 map (FSCwork) or the half2 map (FSCfree) not used in the validation refinement. FSC plots are shown in Supplementary Figs. 9e–11e.

**Structural alignments and analysis**. Superposition of the LCP and cryo-EM structures was carried out in PyMol using a subset of residues corresponding to the TM helices that maintain their relative orientation within each dimer (as measured by interhelical angles). Mainchain atoms for residues 208–272 (TM1-TM2), 494–576 (TM7-9) and 589–612 (TM10) were used for the structural superposition. Overall comparison of the X-ray and cryo-EM structures is further complicated by the fact that there are also small changes at the dimer interface particularly the relative orientation of TM10 and α10 which closely interacts with the adjacent monomer and subtly changes the relative orientation of monomers within each dimer. Consequently for global comparisons we have carried out the superposition across both molecules in the dimer (referred to as the dimer superposition) and for local structural comparisons we have restricted the alignment to the isolated monomers (monomer superposition) using the above residue range. The 'dimer superposition' was used to prepare Fig. 6c, d, Fig. 8a, d, Supplementary Fig. 12a, c-f and Supplementary Movie 1. The monomer superposition was used in the preparation of Fig. 8b, c, Supplementary Fig. 12b, g, h, i.

Electrostatic potential surfaces were calculated using the PDB2PQR server and PyMol with the APBS_2.1 plugin. Structural figures were generated using PyMol or UCSF Chimera.

**MD simulations**. The atomic coordinates of TMEM16K were converted to their Martini CG representation[68,69], and built into membranes following the

MemProtMD pipeline[24]. For the data in Fig. 5a−e, membranes were built using a mixture of 65% POPE, 32% POPC and 3% POPS. To account for the loss of the $Ca^{2+}$ ions, the sidechains of the coordinating Asp and Glu residues were modelled in their neutral form. Simulations were run over multiple µs using a 20 fs time-step. Simulations were run in the NPT ensemble at 323 K with the V-rescale thermostat[70], and 1 bar using semi-isotropic Berendsen pressure coupling[71].

Atomistic simulations were run following conversion of 100 ns CGMD snapshots in a POPC bilayer, where several lipids had been caught mid-scramble. Conversion were carried out using the CG2AT protocol[72] where the original experimentally resolved protein coordinates, including $Ca^{2+}$ ions, were substituted for the post-CG protein coordinates. The Charmm36 forcefield[73] was used to describe the system. Electrostatics were handled using the Particle-Mesh-Ewald method, and a force-switch modifier was applied to the Van der Waals forces. Dispersion corrections were turned off. Simulations were run for 100 ns using 2 fs time steps with Velocity-rescaling temperature coupling at 325 K using a time constant of 0.1 ps and Parrinello−Rahman semi-isotropic pressure coupling of 1 bar with a time constant of 2 ps, before extension to 2.1 µs, using 4 fs time steps with virtual-sites on the protein and lipids[74].

All simulations were run in Gromacs 5.1.2[71]. Images of proteins were made in PyMOL[75] or VMD[76]. Graphs were plotted using Python and Matplotlib.

**Reporting summary**. Further information on research design is available in the Nature Research Reporting Summary linked to this article.

## Data availability

The data that support the findings of this study are available from the corresponding author upon reasonable request. X-ray crystallography data are deposited in the Protein Data Bank (PDB) with the following accession codes 5OC9 and 6R65 for the LCP and vapour diffusion structures, respectively. The cryo-EM density maps and the atomic coordinates have been deposited in the Electron Microscopy Data Bank (EMDB) and in the PDB, respectively, under the following accession codes: EMD-4746/PDB 6R7X, EMD-4747/PDB 6R7Y and EMD-4748/PDB 6R7Z, for 2 mM $Ca^{2+}$, 430 nM $Ca^{2+}$ and $Ca^{2+}$-free forms respectively. The source data underlying Figs. 1b, d, e and f, Fig. 2b−f, Fig. 2h, Supplementary Figs. 1b and 3a−f are provided as a Source Data file.

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

## Acknowledgements

The SGC is a registered charity (number 1097737) that receives funds from AbbVie, Bayer Pharma AG, Boehringer Ingelheim, the Canada Foundation for Innovation, Genome Canada, Janssen, Lilly Canada, Merck KGaA, Merck & Co., Novartis, the Ontario Ministry of Economic Development and Innovation, Pfizer, São Paulo Research Foundation-FAPESP and Takeda, as well as the Innovative Medicines Initiative Joint Undertaking ULTRA-DD grant 115766 and the Wellcome Trust (106169/Z/14/Z). N.J.G. R. and P.T. were supported by the Wellcome Trust (OXION grant no: WT084655MA) and the British Heart Foundation. R.S. is funded by the British Heart Foundation. P.J.S. and R.A.C. were supported by the Wellcome Trust (208361/Z/17/Z) and the BBSRC (BB/ P01948X/1, BB/I019855/1). T.D.N. was supported by an EPSRC Doctoral Training Centre studentship. We thank Diamond Light Source Ltd for access to the macro-molecular crystallography beamlines (BAG mx10619/mx15433) and we thank the I24 beamline staff for their assistance. We acknowledge the Oxford Particle Imaging Centre (OPIC) for providing access to electron microscopes for grid screening. We also acknowledge the Central Oxford Structural Microscopy and Imaging Centre (COSMIC) for providing access to electron microscopes for grid screening and data collection. We acknowledge the use of the UCSF Chimera package[67] from the Resource for Bio-computing, Visualisation, and Informatics at the University of California, San Francisco (supported by NIGMS P41-GM103311). Extended molecular dynamics simulations were carried out using computer time on the ARCHER UK National Supercomputing Service (http://www.archer.ac.uk), provided by HECBioSim, the UK High End Computing Consortium for Biomolecular Simulation (hecbiosim.ac.uk), supported by the EPSRC (EP/L000253/1). We thank all members of the SGC Biotech team, including Claire Strain-Damerell; Kasia Kupinska, Dong Wang and Katie Ellis. We thank all members of the SGC IMP1 group, including Yin Yao Dong and Andrew Quigley. We thank David Eberhardt for help with the electrophysiology experiments. We are grateful to Rod Chalk, Georgina Berridge and Oktawia Borkowska for help with mass spectrometry and Brian Marsden and David Damerell, James Bray, James Crowe and Chris Sluman for bioin-formatics and computer support. We thank Frank von Delft, Tobias Krojer and Beth Maclean for assistance with crystallography infrastructure. We thank Owen Vickery for implementation of the lipid virtual sites and Mark Sansom for discussions. We thank Errin Johnson and Adam Costin (COSMIC) for assistance with grid screening and data collection. We thank Andrea Nemeth for critical reading of the manuscript.

## Author contributions

S.R.B., A.C.W.P. and M.E.F. contributed equally to this work and should be regarded as joint first authors. N.J.G.R., C.M.T., R.A.C., T.D.N., J.C.C. and L.F.S. contributed equally to this work and should be regarded as joint second authors. R.S., P.J.S., J.T.H., P.T. and A.A. jointly co-supervised various different aspects of this work with overall supervision by E.P. C. S.R.B., A.C.W.P., R.S., P.T., P.J.S., A.A. and E.P.C. designed the project. S.R.B. optimised and prepared protein samples, obtained crystals and prepared cryo-EM grids with C.A.S., assisted by A.T. and A.C. Q.W. assisted with lipid binding studies. S.R.B., A.C.W.P. and E. P.C. screened crystals and collected X-ray data sets. S.R.B., C.A.S., J.T.H., A.C.W.P. and E. P.C. screened grids and collected cryo-EM data. S.R.B. and A.C.W.P. determined the structures. L.S. and S.M.M.M. performed pilot studies for protein production and produced baculovirus infected cells. Work by L.S., and S.M.M.M. was supervised by N.A.B-B. J.D.L. was involved in pilot studies for protein production. N.J.G.R., L.F.S, C.M.T. and P.T. performed and analysed patch-clamp experiments. N.J.G.R., J.C.C., L.F.S., and P.T. per-formed and analysed the immunocytochemistry. M.E.F. performed the lipid scramblase and bulk ion channel experiments, supervised by A.A. S.R.B attempted single channel e-physiology experiments (not shown as the results were inconclusive), supervised by R.S. MD simulations were designed, performed and analysed by R.A.C., T.D.N. and P.J.S. E.P.C. was responsible for the overall project design, data collection, supervision and manage-ment. S.R.B., A.C.W.P., M.E.F., N.J.G.R., C.M.T., R.A.C., T.D.N., J.C.C., L.F.S., R.S., P.J.S., J. T.H., P.T., A.A. and E.P.C. analysed the data and prepared the manuscript.

## Additional information

**Competing interests:** The authors declare no competing interests.

