## [Peer Review File · Nature Communications]

Reviewers' comments:

Reviewer #1 (Remarks to the Author):

This manuscript presents several structures of the human TMEM16K protein: a moderate (3.4Å) resolution X-ray crystal structure in the presence of 100 mM Ca, as well as two low-resolution cryo-EM structures in the absence (5.3Å) and presence of 500 nM Ca (4.5Å). While the structure determination is sound, my enthusiasm for this manuscript is limited for several reasons.

The functional characterization of this protein is in its infancy. Its precise physiological role is far from known, let alone its subcellular localization. The authors attempt to address the latter, but overexpression in HEK cells is not a convincing way to demonstrate physiological subcellular localization. Therefore, the premise for solving a structure of TMEM16K is not well established. In other words, what particular functional mechanism will be elucidated by the structure?

Although the authors show that TMEM16K does have some Ca-dependent scramblase activity, others have shown that TMEM16K does not have any scramblase activity (Suzuki et al, JBC 2013). This discrepancy is not discussed. It is also unclear how relevant the scramblase activity rate determined with the dithionite assay is to a potential physiological rate – since a physiological rate is not known. Likewise, how does the apparent EC50 for Ca in the dithionite assay compare to a physiological EC50? These points would need to be addressed in order to claim that the structure of a physiologically-relevant complex is being determined and analyzed. In other words, these comparisons would allow the authors to establish whether or not accessory proteins, post-translational modifications, or specific lipids are required to recapitulate physiological function.

Finally, even if one accepts that TMEM16K is a bona fide Ca-dependent lipid scramblase, the structures do little to explain the mechanism for this putative feature. While the X-ray structure in the presence of Ca had an open groove, the cryo-EM structure in the presence of 500 nM Ca had a closed groove, which was quite similar to what was observed in the Ca-free cryo-EM structure (and, by the way, maps should be shown to make a much more convincing case for the claimed structural changes at this low of a resolution). Comparing these structures does little to explain how Ca-binding accelerates the scramblase rate. It is also not discussed why the authors even expect to see Ca-bound in the 500 nM cryo-EM structure. Since the grids were prepared with 65 µM protein, less than 1 % of the protein molecules would be expected to bind Ca, and substantially less would even be fully occupied since the expected stoichiometry is 3.5:1. Unless the authors claim that Ca could be co-purified, then this calls into question the accuracy with which they are able to identify bound Ca in their maps.

Reviewer #2 (Remarks to the Author):

This is an excellent manuscript showing strong biochemical and structural data that TMEM16K is a phospholipid scramblase. The TMEM16 proteins have recently attracted a great deal of attention because of their broad physiological significance and their disease relevance. This paper makes a significant advance by providing a wealth of data on the structure and function of this member of the TMEM16 family. This is important because TMEM16K, which is ubiquitously expressed, has received relatively little attention and its function and structure have not been solved. Despite my general enthusiasm for this study, the paper has a number of serious flaws that must be resolved.

Major points:

(1) The confocal data that TMEM16K is located in the ER (Fig. 1a-b) is weak for several reasons. First, Con-A is not a specific probe for the ER. While Con-A staining may be more intense for high mannose oligosaccharides on ER proteins, Con-A also labels plasma membrane, secretory granules, and Golgi. The authors should use a more specific ER-resident protein (for example, Sec61b). HEK cells are not a good choice for subcellular structure – the authors might have better success with COS7 cells. Co-expression of Sec61b and TMEM16K should reveal very nice ER tubules and sheets. Second, the spatial resolution of the technique is inadequate. ER tubules are ~100 nm thick, but the plots cover a distance of 20 – 30 µm and there is no information how Con-

A and TMEM16K colocalize on a submicron level. A quantitative evaluation of colocalization is required on a pixel-by-pixel level. Also, images need to be collected in the z-dimension and deconvolved because one cannot tell if superimposed red and green pixels are located in different planes. Third, the fluorescent images are difficult to interpret without some reference structures – nucleus, cytoskeleton, etc. Finally, overexpressed proteins frequently get stuck in the ER, so while the bulk of the protein may be stuck in the ER, there may be a physiologically-relevant fraction in other locations. It would be nice to show its location in native cells, although availability of antibodies may preclude this.

(2) It is very puzzling that the conformation of the cryo-EM structure in 500 nM is so different from the X-ray structure in high Ca, especially because both Ca binding sites are similarly occupied in both structures. This raises the question whether the “open” conformation of the protein seen in the X-ray structure is real or whether it is an artifact of the different procedures used to create the X-ray and the cryo-EM models. Because scrambling by TMEM16K is 300-fold slower than the fungal homologs, it seems possible that TMEM16K is not a scramblase but rather is an ion channel and that the X-ray structure is anomalous and misleading. The X-ray structure shows a 4th Ca ion bound at the N-terminal end of TM10, maybe coordinated by the carbonyl oxygen of E586, but the authors do not say anything about this. This ion is missing in both cryo structures. While this site is luminal and unlikely to be physiologically relevant to Ca-dependent gating, it is possible that the difference in structure of the cryo and X-ray structures is related to binding of this ion. These issues must be explained. For me to be convinced, it will be necessary to see a cryo-EM structure in high Ca that has the same conformation as the X-ray structure.

(3) It is also strange that CGMD shows no scrambling of the cryo-EM structure despite the fact that the experimental data show that scrambling is quite robust in the absence of Ca (Ca-free rate constant is 25% that of the Ca-stimulated system). The Ca-stimulated system shows dozens of transitions (Fig. 4C-D), while the Ca-free system does not show even a single lipid entering the groove from either side (Suppl. Fig. 8i). In that regard, it seems that the authors have been a little disingenuous in showing only a 2- μ sec simulation for the Ca-free state while they show 10 μ sec for the Ca-stimulated state.

(4) The authors suggest that the Asp615Asn mutation may explain the SCAR10 ataxia because it is involved in Ca binding which could affect channel structure or gating. This needs verification because there are no published experimental data showing that this amino acid in any TMEM16 protein affects channel function. The authors should test this mutant in the liposome assay.

(5) The proteoliposome assay shows TMEM16K scrambles lipids at a rate 300-fold slower than the fungal TMEM16s. One explanation the authors propose to explain this difference is “sensitivity to membrane composition”. This suggests that the experiments on TMEM16K and the fungal proteins were carried out under different conditions. Because both measurements were presumably performed by the Accardi lab, this explanation should be tested in side-by-side experiments. Another explanation the authors propose is “small molecules, proteins, or lipid cofactors”. This needs better justification. Where do the authors imagine these cofactors coming from? An explanation that the authors do not offer is that the TMEM16K protein is partly denatured or inactive. In this regard, the authors should explain and show data for how many proteins are incorporated per liposome.

Minor issues:

(6) Although the TMEM16 nomenclature is favored by many investigators in the field, the ANO terminology is the preferred name approved by the nomenclature committees and should be cross-referenced.

(7) Page 5. Line 2. Should include reference #8.

(8) Fig. 1f. By my eye, I cannot tell the difference between the color used for TMEM16K 0.5 mM Ca and CLC-ec1.

(9) The statements about ion channel activity are disconcerting. What exactly does “inconclusive” mean? Considering my concerns expressed in #2 above, it seems necessary to have a clear

understanding about whether this is a channel or not.

(10) Figure 4c,d shows that lipids move in both directions, however, it is not clear how this happens. Do the lipids pass one another going different directions or is there a concerted alternation in the direction of conduction through the pathway?

(11) Fig. 4b, the gold headgroups look grey in my copy.

(12) Fig. 4e,f. In the atomistic MD, it does not seem that any of the lipids make a full translocation from one side to the other, unless I misunderstand the figure. The statement on P8 "We also observed lipid scrambling upon conversion of intermediate states from the CG simulations to an atomistic description (Fig 4d), with lipids sampling the full translocation pathway." is ambiguous. I don't know what the authors mean by "scrambling upon conversion of intermediate states." The authors seem to equate scrambling with occupancy of the cleft with lipids rather than actual translocation of a single lipid from one side to the other. Please clarify.

(13) Page 8. The statement "nhTMEM16 is one of the few proteins to reveal lipid scrambling, within the >3500 membrane proteins in MemProtMD" is inscrutable. Does this mean that all 3500 proteins in MemProtMD have been examined for scrambling in silico or in wet experiments?

(14) Typo: Supplementary Figure 3d. mTMEM16A is not 5OCB.

(15) Page 12. These movements recapitulate the changes seen in aTMEM16 35 should also cite ref. 8.

Reviewer #3 (Remarks to the Author):

Bushell et al. have carried out a well-designed study, elucidating the structure of TMEM16K. Interestingly, they report that TMEM16K is an ER-resident lipid scramblase. Both high-resolution Xray structure and medium-resolution CryoEM maps are giving insight into the possible conformational dynamics. In addition, molecular dynamics simulation is showing a possible scrambling mechanism. The results are an important step towards unraveling the lipid flipping in ER and elucidating its role in some diseases such as ataxia.

Overall, the manuscript is well written, and the experiments are soundly presented. The findings are novel and are definitely of interest to others in the community and the wider field. Thus, I support the publication of this manuscript in Nature Communications upon addressing the comments below.

Remarks to the author:

1. A surprising issue about the manuscript is the general neglect (or minimal discussion) about the importance of lipid role in conformational change. Considering the lipid scrambling activity and significant lipid remodeling upon conformational change makes even more unexpected not to investigate the lipid role thoroughly.

a. The crystal structure has been obtained in LCP crystallization and it has been mentioned in the manuscript that there are several MAG lipids in the dimer interface; it would be interesting if there were any indication of lipid binding into the grooves.

b. Also, previous study (Malvezzi et al, ref29) showed that lipid composition of membrane plays a vital role in the lipid scramblase activity of aTMEM16, which would probably be the case for TMEM16K also and at the same time the selectivity of lipid scrambling by TMEM16K is not clear. Both aspects deserve a more detailed discussion.

c. In the CG simulation, as there are several scrambling events, it would be very interesting to see if there is a preference for any of the lipid used in the simulation.

2. In silico scrambling analysis has shown interesting results that justifies a more detailed description:

a. In the CG MD simulation there are enough scrambling events in both directions to make a good estimation of kinetic rate of lipid scrambling by TMEM16K.

b. There is an interesting observation in the CG simulation (Fig 4c,d). In the first 2 μ s it seems

that flipping is a directional process (i.e lipids are only flipping from the cytoplasmic to luminal leaflet). Is there any explanation for this observation?

c. Presenting the same quantification of lipid flipping for the atomistic simulation (like Fig 4c,d) would be nice.

d. There is no description of the residues involved in the flipping process. Also a comparison with the findings of previous studies (Bethel & Grabe PNAS and Jiang et al eLife) would be beneficial.

3. Atomistic simulations have been started after a short CG simulation rather than starting directly from the structure, is there any reason for this protocol? The electrostatic interactions are poorly described in the Martini force field; therefore, the Ca binding site probably would not be accurately preserved during the CG simulation. Also, as the atomistic simulations have been performed in a homogenous bilayer (POPC only), they do not require equilibrated mixing of different lipid species.

4. As the experimental results about the ion channel activity were inconclusive, it would be interesting to know if MD simulation is showing any indication about ion transport? Or whether the grooves serve as the pathway for ion conductance also (as shown for nhTMEM16, Jiang et al 2016 ref 8).

5. In Fig 6c, considering the medium resolution of the EM maps how much the side chains' positioning is reliable? Addition of actual maps makes the comparison trustworthy.

6. References need a careful check and editing. For example, reference number 31, 33 and 39 (Malvezzi et al) are identical!

Reviewer #4 (Remarks to the Author):

In this interesting and important contribution by Bushnell and colleagues, the authors present X-ray and cryo-EM structures of the ER-resident human lipid scramblase TMEM16K. X-ray structures of lipidic cubic phase and hanging drop crystals at 3.4 Å and 4.0 Å reveal that TMEM16K adopts the canonical TMEM175-family butterfly fold. The putative TMEM175 scramblase groove is open in the structure and several densities were resolved within the groove that are assigned as lipids. The role of the groove in facilitating lipid scramblase activity is supported by coarse grained MD simulations that indicate that lipid headgroups can use the groove to cross the membrane in either direction.

Anomalous diffraction of the LCP crystals identified several bound Ca²⁺ ions. Two ions are present at the two Ca²⁺-binding site formed by TM6 to TM8. Ions were also identified at this binding site in the Ca²⁺-bound mTMEM16A channel cryo-EM structure. Ions were also resolved at the TM10/a10 junction near the subunit interface. Inspection of the mTMEM16A cryo-EM density map suggested that a Ca²⁺ ion may also be present and that the TM10/a10 binding site may be conserved. A fourth Ca²⁺ ion was found at the dimer interface between TM10 of each subunit.

Using cryo-EM analysis of TMEM16K vitrified in 10 mM EGTA or 500 nM Ca²⁺, two additional conformations were resolved at 5.3Å and 4.5Å. In the 500 nM Ca²⁺ condition, TMEM16K adopts a conformation in which the scramblase groove is narrowed near the ER side of the membrane due a movement of TM3 and TM4. The narrowing of the groove is accompanied by a large movement of the cytoplasmic domain. MD simulations support the hypothesis that this conformation cannot facilitate lipid scramblase activity. The scramblase groove is also closed in the Ca²⁺-free structure, though the conformation is distinct from either of the Ca²⁺-bound conformations. In the absence of Ca²⁺, the two Ca²⁺ binding site expands and TM10/a10 site undergoes a rearrangement. Similar conformational changes were also detected in structure of a fungal TMEM16 scramblase in the presence and absence of Ca²⁺.

In addition to its important contribution to understanding TMEM16 scramblases, this work is an excellent example of combining X-ray crystallography and cryo-EM to analyze the structure of a dynamic molecule at varying resolutions. I have several suggestions that may strengthen the presentation of the data.

1. The authors state that no currents could be recorded from cells transfected with TMEM16K, indicating that TMEM16K is exclusively a scramblase. As Figure 1 shows that TMEM16K is primarily localized to the ER and currents are being recorded from the plasma membrane, it is possible that the lack of channel activity is due to a low expression level in the plasma membrane. As the

authors further state that bilayer and flux-based assays are inconclusive, it would be better to refrain from claiming that TMEM16K is not a channel without more definitive data. The scramblase activity appears to be robust and is the focus of this work.

2. Extensive 2D classification (5-6 rounds) was used to filter the cryo-EM image data prior to further classification steps. Such extensive classification can exclude both "bad" and "good" particles. As the Ca²⁺-bound cryo-EM structure is quite distinct from the x-ray structures, is it possible that particles belonging to alternative conformations are being excluded during 2D or 3D classification. A thorough understanding of any conformational heterogeneity present in the cryo-EM data would be very helpful for starting to understand why such different conformations were obtained by X-ray crystallography and by cryo-EM. If the cryo-EM data set does indeed contain only a single conformation, a more complete discussion should be included as to why the protein adopted such distinct conformational states under similar conditions. Is the presence of the extra Ca²⁺ binding site on the TM10-TM10 axis contributing to the open conformation?

3. In Figure 5, HOLE is used to calculate the volume of the scramblase groove in the open and closed configurations. At 4.5Å resolution the side-chain densities are unlikely to be well enough resolved for accurately modelling. As differences in side-chain positions can dramatically alter the shape of a cavity, a 4.5Å structure is too low for such calculations. It would be better to remove the HOLE calculation and the side-chains from Figure 5 and focus on the changes in the positions of TM3 and TM4. A similar analysis of the Ca²⁺-free state would also be informative for the reader.

4. On a related note, how were the MD analyses of the Ca²⁺-bound closed conformation performed? Is the accuracy of the model based on a 4.5 Å structure sufficient for the simulation?

Response to reviewers comment for Bushell et al., TMEM16K manuscript

We thank all the reviewers for their helpful comments and suggestions. Several of these suggestions have prompted us to perform additional experiments which led to improvements in the understanding of how TMEM16K functions. As the reviewers raised similar points, we briefly outline the steps we took to address the main concerns, followed by detailed point-by-point responses to the individual points raised.

i. The data supporting the cellular localisation of TMEM16K was not sufficient to conclude that TMEM16K is an ER-localized protein.

To strengthen our localisation data, we performed additional experiments using updated experimental methodologies.

- a. We probed the cellular localization of endogenous TMEM16K in the osteosarcoma cell line U-2 OS (referred hereafter as U2OS), which has high levels of TMEM16K transcript and protein. We observed TMEM16K immunostaining consistent with ER localization in untransfected U2OS cells (Fig. 1,c,d) that was not present in either HEK293T or COS7 cells (Supplementary Fig. 1c).
- b. We show that TMEM16K localises to the ER using COS7 cells heterologously expressing a tagged form of human TMEM16K (FLAG-His₁₀-TMEM16K) and co-staining with two independent ER markers; the ER-resident chaperone calnexin (CNX)(Fig. 1a,b) and the ubiquitin ligase Hrd1 (Supplementary Fig. 1a,b). Fig. 1a,b and Supplementary. Fig. 1a,b illustrate representative examples of cells stained for these targets. As can be appreciated from visual inspection of these images, there is significant colocalisation between TMEM16K and both of these ER markers.
- c. We quantified the extent of co-localisation using the Coloc2 tool in FIJI.
- d. Since our data does not rule out the possibility that TMEM16K can also localize to other compartments, we now rephrased our conclusions to state that “TMEM16K is capable of ER localization...” (e.g. Page 6, Ln 2) or “...mainly resident in the ER...” (Page 9, Ln 2).

ii. The reviewers were concerned by the slow scrambling rate of TMEM16K and by the finding that our cryoEM structures show a closed pathway, despite the presence of Ca²⁺ bound to the intramembrane sites. Therefore, they raised the possibility that TMEM16K might be a channel, and the crystal structure reflects a rarely-visited conformation.

To address these concerns we performed several additional functional and structural experiments:

- a. We show that TMEM16K mediates fast lipid scrambling when reconstituted in thinner membranes, which more closely mimic its native ER membrane environment (Fig. 2c,d).
- b. When reconstituted in these thinner membranes, TMEM16K also mediates non-selective channel activity (Fig. 2g,h).

These findings show that TMEM16K recapitulates the key functional properties of other TMEM16-type scramblases; the fungal afTMEM16 and nhTMEM16 as well as of the mammalian TMEM16F and TMEM16E. Thus, our findings support the notion that TMEM16K is a *bona fide* TMEM16-type scramblase.

- c. We determined an additional cryoEM structure of TMEM16K in the presence of 2 mM Ca²⁺. We found that in these conditions the lipid pathway is closed (Fig. 6f,h). While this finding is somewhat unexpected, it is not unlike what was recently reported for TMEM16F, whose pathway is closed in the presence of saturating Ca²⁺ (Alvadia et al., 2019, eLife, e44365). Further, the nhTMEM16 scramblase adopts an open conformation in the absence of Ca²⁺ (Kalienkova et al., 2019. eLife, e44364), suggesting that factors other than Ca²⁺ also control the conformational state of the pathway of TMEM16 scramblases.

- ✎ Finally, we note that the scrambling assay is a macroscopic measurement of lipid transport. Thus, we cannot estimate the absolute open probability of the groove in any specific condition; which is reflected in the conformation determined in the structural experiments. The assay can only inform on the relative changes in probability due to the various manipulations. It is thus possible that the “maximal” open probability of the TMEM16K scramblase is $\ll 1$, so that the intermediate Ca^{2+} -bound closed pathway conformation is the most likely to be identified in these structural experiments. This is similar to the situation in electrophysiology, where macroscopic current recording provide estimate of the relative P_o , while determination of the absolute P_o necessitates single channel recordings or noise analysis. This issue is now discussed in Page. 18, Ins 6 – 19.

Our finding that the TMEM16K pathway can adopt both open and closed conformations that resemble those seen in fungal TMEM16 scramblases (Falzone et al., eLife, e43229, 2019; Kalienkova et al., eLife, e44364, 2019) shows that mammalian TMEM16 scramblases can undergo similar rearrangements, which had not been previously demonstrated (Alvadia et al., eLife, e44365, 2019). It is worth noting that our findings, together with those recently reported for nhTMEM16 (Kalienkova et al., eLife, e44364, 2019; Alvadia et al., eLife, e44365, 2019) suggest that Ca^{2+} is not the only determinant of activity of the TMEM16 scramblases. Rather, our new data suggests that other factors, such as the lipid composition and/or physico-chemical properties of the surrounding lipid membrane, or the composition of the detergent micelles or LCP in meso phase in crystals, modulate the relative depth of the energetic minima corresponding to the different conformations of the pathway. This is now discussed on Page 18, Ins 15-19.

Detailed responses

Reviewer #1

1. The functional characterization of this protein is in its infancy.

We agree with the reviewer that the functional characterization of TMEM16K was incomplete. Accordingly, we performed additional experiments to characterize its lipid selectivity, modulation by bilayer properties and channel activity.

- We found that TMEM16K is a poorly selective scramblase: PE, PC and PG are scrambled at comparable rates, while PS is transported ~ 3 -fold slower (Fig. 2e,f, Supplementary Figure 3c,d).
- The scramblase activity of TMEM16K is exquisitely modulated by membrane thickness, it is high in thin membranes and low in thicker ones (Fig. 2c-d, Supplementary Figure 3a-d).
- Although we were initially unable to detect ion channel activity, when we tested whether incorporation in thinner membranes also increased the channel activity of TMEM16K, we found that this is the case (Fig. 2g-h). So channel activity, as well as scramblase activity, is much improved in thinner membranes.

These results show that TMEM16K recapitulates all the key properties of fungal and mammalian TMEM16-type scramblases.

2. Its precise physiological role is far from known, let alone its subcellular localization.

We agree with the reviewer that the physiological role of TMEM16K is far from known. In the present manuscript we show that TMEM16K is a dual-function scramblase/non-selective channel, that it is capable of ER localization and we elucidate the major conformational rearrangements that underlie its activation. These findings provide the functional and structural framework necessary to address the question of what its physiological role is.

3. The authors attempt to address the latter, but overexpression in HEK cells is not a convincing way to demonstrate physiological subcellular localization.

We agree that our data was insufficient to conclude that TMEM16K was localized in the ER. We now present additional evidence both in cell lines that endogenously express TMEM16K and in overexpression systems supporting our conclusion. Please see point (i) above and Points 1-5 in our response to Reviewer #2 below.

4. Although the authors show that TMEM16K does have some Ca-dependent scramblase activity, others have shown that TMEM16K does not have any scramblase activity (Suzuki et al, JBC 2013). This discrepancy is not discussed.

We apologise for this omission. The Suzuki et al., JBC, 13305-16. 2013 work looked at scrambling at the plasma membrane (PM), so it could not discriminate between lack of scrambling activity at the PM and intracellular localization. Our findings suggest that their conclusion that TMEM16K does not scramble at the PM was due to its intracellular localization. This issue also occurred with another intracellularly localized scramblase TMEM16E, which showed no activity in the 2013 Suzuki et al work, but was later shown to be a scramblase (Di Zanni et al., Cell Mol Life Sci, 2017; Whitlock et al., JGP, 2018). To discuss this issue we added the following statement on page 6: "These results are in agreement with reports suggesting that no scramblase activity was detected at the PM when TMEM16K was expressed in TMEM16F^{-/-} mouse cells¹¹" (Ref 11: Suzuki, J. et al. Calcium-dependent phospholipid scramblase activity of TMEM16 protein family members. J Biol Chem 288, 13305-13316, 2013).

4. It is also unclear how relevant the scramblase activity rate determined with the dithionite assay is to a potential physiological rate – since a physiological rate is not known.

To the best of our knowledge, quantitative measurements of scrambling rates in cells have not been performed for any scramblase as the kinetic resolution of the available tools (i.e. Annexin V binding, internalization of fluorescently tagged lipids, LactC2 binding) is limited by factors other than lipid transport, mainly by the kinetics of binding or internalization of the tagged lipids. Further, these measurements are limited to the plasma membrane of cells, while TMEM16K primarily localized to intracellular compartments. We would very much appreciate it if the reviewer could recommend methods to perform these important experiments. We note that the estimates of the lipid scrambling rates by TMEM16 proteins obtained with the dithionite assay (Malvezzi et al., Nat. Comms., 4, 2367 (2013), Malvezzi, et al., PNAS, 115,E7033-E7042,) are in agreement with those obtained using other *in vitro* methods (Watanabe et al., PNAS, 115, 3066-3071), attesting to the robustness of this assay. Finally, we now show that the rate of lipid scrambling by TMEM16K is enhanced in thinner membranes. Since the ER membrane is thinner than the PM, we propose that the sensitivity of TMEM16K to the membrane environment might be of physiological relevance. Further work will be needed to test this hypothesis. The results we present are therefore the best available estimate of scrambling rates for TMEM16K.

5. Likewise, how does the apparent EC50 for Ca in the dithionite assay compare to a physiological EC50?

We did not measure the EC₅₀ for Ca²⁺ in the present manuscript. In cases where such comparisons were possible, such as the TMEM16A channel (Terashima et al., PNAS, 2013) and the TMEM16F scramblase (Alvadia et al., eLife, 2019) the values estimated using liposome assays recapitulate well those measured using cell-based electrophysiological measurements.

6. These points would need to be addressed in order to claim that the structure of a physiologically-relevant complex is being determined and analyzed. In other words, these comparisons would allow the authors to establish whether or not accessory proteins, post-translational modifications, or specific lipids are required to recapitulate physiological function.

To test whether the activity of TMEM16K depends on accessory proteins, post-translational modifications or specific lipids we compared the activity of protein purified from insect (sf9) and mammalian (HEK293) cells, to determine if components not found in sf9 cells but present in the mammalian HEK cells are necessary. We found that the activity was similar for protein from both cell types (Supplementary Figure 3), suggesting this is not the case.

7. Finally, even if one accepts that TMEM16K is a bona fide Ca-dependent lipid scramblase, the structures do little to explain the mechanism for this putative feature. While the X-ray structure in the presence of Ca had an open groove, the cryo-EM structure in the presence of 500 nM Ca had a closed groove, which was quite similar to what was observed in the Ca-free cryo-EM structure (and, by the way, maps should be shown to make a much more convincing case for the claimed structural changes at this low of a resolution).

As discussed above (see point (ii)), our finding that the TMEM16K groove is closed in the presence of Ca^{2+} is consistent with the idea that the activation mechanism of the TMEM16 scramblases is not a simple closed $\leftarrow\rightarrow$ open equilibrium, but rather that it entails multiple intermediate states. Our observation that the lipid chain length is important for optimal activity, together with the observation that lipids bind in the dimer cavity, suggests that TMEM16K transitions from open to closed states may also depend on factors such as the lipid environment, in addition to Ca^{2+} . Such multi-state mechanisms were recently proposed based on the structures of the fungal afTMEM16 (Falzone et al., eLife, 2019) and nhTMEM16 (Kalienkova et al., eLife, 2019). Additionally, the different conformations we identify recapitulate those reported for other TMEM16 scramblases (Falzone et al., eLife, 2019; Kalienkova et al., eLife, 2019; Alviafia et al., eLife, 2019). Our finding that bilayer thickness is a critical regulator of TMEM16K scrambling activity (Fig. 2) suggests that Ca^{2+} may not be the sole regulator of the transition between these states. This is now discussed on page 18 and in figure 8.

8. It is also not discussed why the authors even expect to see Ca-bound in the 500 nM cryo-EM structure.

The EC_{50} for Ca^{2+} for most known TMEM16 channels and scramblases is reported to be in the low micromolar range (1 μM or 7 μM for TMEM16F (Alviafia, et al., eLIFE, 8, e44365, 2019; Ye, et al., PNAS, 115, E1667–E1674, 2018); 0.4 μM for TMEM16A (Yang et al., Nature, 455, 1210-1215, 2008)). Given the high conservation of the Ca^{2+} -coordinating residues, we hypothesize that TMEM16K is also going to be in this range, or perhaps it may have even higher affinity (see below). Thus, at 500 nM Ca^{2+} we expect the sites to be occupied a significant fraction of the time, which is consistent with our observation.

9. Since the grids were prepared with 65 μM protein, less than 1 % of the protein molecules would be expected to bind Ca, and substantially less would even be fully occupied since the expected stoichiometry is 3.5:1. Unless the authors claim that Ca could be co-purified, then this calls into question the accuracy with which they are able to identify bound Ca in their maps.

We apologise to the reviewer for not clearly explaining our experimental conditions. The sample described as 500 nM was purified without the addition of either Ca^{2+} or EGTA, so the Ca^{2+} present in the sample was obtained from the cells and the purification buffers. To estimate the concentration of Ca^{2+} in our samples, we took the following steps.

- We measured the background concentration of Ca^{2+} in our size exclusion chromatography buffers and found that it was (430 ± 73) nM ($n=3$, s.e.m. used to calculate the error), so we now refer to the sample prepared without addition of Ca^{2+} or EGTA as the 430 nM Ca^{2+} sample (page 13).
- We measured the Ca^{2+} concentration of a solution containing 1 mg/ml protein in buffer to which no additional Ca^{2+} had been added, from which we removed the protein using a concentrator with a molecular weight cut off of 10 kDa. The eluate contained (1020 ± 253) nM Ca^{2+} (s.e.m, $n=3$).

- We tested the concentration of Ca²⁺ directly in protein samples without denaturing the protein with heat. When the protein was denatured the Ca²⁺ concentration was (37.2 ± 10.7) μM (2 different experiments, 2 different protein preps, 3 samples of each, i.e. n=6). The protein concentration was ~13 μM, so with three Ca²⁺ ions per monomer, we would expect 39 μM Ca²⁺, similar to the concentration measured.
- Interestingly, when TMEM16K was not denatured, the measured free Ca²⁺ concentration was much lower, (2191 ± 243) nM (s.e.m., n=3), suggesting that TMEM16K can compete with the dye for Ca²⁺ ions. Since the dye has a K_d of 190 nM for Ca²⁺, this suggests that the affinity of TMEM16K for Ca²⁺ is higher than 200 nM, consistent with our finding that at 500 nM the sites are occupied. However, an accurate estimate of the affinity for Ca²⁺ will require further work.

These points are now briefly described in the Results section (Page 13, lns 20-24) and more thoroughly in the methods section (Page 26).

Reviewer #2

We thank reviewer #2 for their positive comments.

1. The confocal data that TMEM16K is located in the ER (Fig. 1a-b) is weak for several reasons.

We thank the reviewer for pointing this out. As discussed above (see point (i)) we have now carried out extensive additional localization experiments on both native and overexpressed TMEM16K and show that it is capable of ER localization.

2. Con-A is not a specific probe for the ER. While Con-A staining may be more intense for high mannose oligosaccharides on ER proteins, Con-A also labels plasma membrane, secretory granules, and Golgi. The authors should use a more specific ER-resident protein (for example, Sec61b). HEK cells are not a good choice for subcellular structure – the authors might have better success with COS7 cells. Co-expression of Sec61b and TMEM16K should reveal very nice ER tubules and sheets. Second, the spatial resolution of the technique is inadequate. ER tubules are ~100 nm thick, but the plots cover a distance of 20 – 30 μm and there is no information how Con-A and TMEM16K colocalize on a submicron level.

We have updated the experimental methodology and now demonstrate ER localisation using COS7 cells heterologously expressing a tagged form of human TMEM16K (FLAG-His₁₀-TMEM16K) through a viral construct, which we have co-stained with two independent ER markers; the ER-resident chaperone calnexin (CNX) and the ubiquitin ligase Hrd1. Figure 1 and Suppl. Fig. 1 illustrate representative examples of cells stained for these targets. As can be appreciated from visual inspection of these images, there is significant colocalisation between TMEM16K and both of these ER markers. As outlined below, we have quantified the extent of co-localisation using the Coloc2 tool in FIJI.

3. A quantitative evaluation of colocalization is required on a pixel-by-pixel level. Also, images need to be collected in the z-dimension and deconvolved because one cannot tell if superimposed red and green pixels are located in different planes.

We have now taken each image from a single plane with a pinhole size 0.70 AU. The extent of co-localisation of TMEM16K with ER markers was quantitatively assessed by Manders' overlap coefficient (MOC) and Pearson's correlation coefficient (PCC) using Fiji software with the Coloc2 plugin^{1,2}. Two different Manders' coefficient values were obtained (M1 and M2); these quantify the independent contributions of two selected channels to the pixels of interest. M1 represents the fraction of the green channel (TMEM16K) in regions containing red signal (ER marker) while M2 conveys the fraction of the red channel in regions containing green signals. Pearson's coefficients were also calculated as

a measure of pixel-by-pixel cross-correlation coefficients. MOC and PCC values range from 0 (no correlation) to 1 (complete overlap).

4. Third, the fluorescent images are difficult to interpret without some reference structures – nucleus, cytoskeleton, etc.

We have followed the referee's recommendation and stained nuclei using 4',6-diamidino-2-phenylindole (DAPI) during immunostaining.

5. Finally, overexpressed proteins frequently get stuck in the ER, so while the bulk of the protein may be stuck in the ER, there may be a physiologically-relevant fraction in other locations. It would be nice to show its location in native cells, although availability of antibodies may preclude this.

To address this important point, we searched for cell lines with higher levels of TMEM16K transcript (CCLE-Broad Institute, <https://portals.broadinstitute.org/ccle/page?gene=ANO10>) and protein (PaxDb, <https://pax-db.org/protein/1847340>, Geiger et al. MCP 2012-PMID 22278370) levels. In doing so, we identified the osteosarcoma cell line U-2 OS (referred hereafter as U2OS) as a potential candidate to demonstrate localisation of native TMEM16K, given its expression levels were near the upper quartile of all expressed proteins. Using a commercially available TMEM16K antibody (Sigma, HPA051569, rabbit polyclonal, 1:100), we observed TMEM16K immunostaining consistent with ER localization in untransfected U2OS cells that was not present in either HEK293-T or COS7 cells (Supplementary Figure 1c), providing indirect evidence for the specificity of the TMEM16K antibody. We also performed Western blots with extracts from HEK293 cells that were either transduced with baculovirus for expression of TMEM16K, virus with a control unrelated protein or no virus. Only the cells expressing TMEM16K gave bands in the Western Blots, further confirming that the antibody we used is specific for TMEM16K (Supplementary Figure 1d). In U2OS cells, the extent of co-localisation between TMEM16K and the specific ER marker KDEL was quantitatively determined and comparable to that observed in COS7 heterologously expressing TMEM16K. Collectively, these new data more clearly support our claim that TMEM16K is localised to the ER membrane.

Overall we believe that this extensive localisation data, in particular the data on endogenous expression of TMEM16K in U-2 OS cells, together with the lack of channel activity at the PM (our data) or scramblase activity at the PM (Suzuki et al., *J. Biol. Chem.* 288, 13305-13316 (2013)) indicates that TMEM16K is mainly resident in the ER. We have adapted the text to say that TMEM16K is “capable of ER localisation” or “mainly localised in the ER”.

6. It is very puzzling that the conformation of the cryo-EM structure in 500 nM is so different from the X-ray structure in high Ca, especially because both Ca binding sites are similarly occupied in both structures. This raises the question whether the “open” conformation of the protein seen in the X-ray structure is real or whether it is an artefact of the different procedures used to create the X-ray and the cryo-EM models.

We agree with the reviewer that the different conformation adopted by the TMEM16K pathway between the X-ray and cryoEM structures is puzzling. However, as discussed above (see point (ii)), there is a growing body of evidence suggesting that the lipid permeation pathway of the TMEM16 scramblases can adopt multiple conformations (Lee et al., *Nat Comms*, 2018; Falzone et al., *eLife*, 2019; Kalienkova et al., *eLife*, 2019; Alvadia et al., *eLife*, 2019), rather than a single open and closed one. Indeed, similar results were recently reported for the mammalian TMEM16F, which is closed in the presence of Ca^{2+} (Alvadia et al., *eLife* 2019), and the fungal nhTMEM16, which is open even in the absence of Ca^{2+} (Kalienkova et al., *eLife*, 2019). We have shown that loss of Ca^{2+} binding is not sufficient to induce changes from the open to closed states. Rather we find that other factors are involved, such as the lipid composition of the membrane. We propose a 4-state model, in which we have to date observed three of the four available states. We see open and closed states with Ca^{2+} ; we also see a closed state with Ca^{2+} but we have not yet observed the open groove, Ca^{2+} free state (Fig. 8). However, for nhTMEM16 a Ca^{2+} free open groove conformation has been observed (Kalienkova, *eLIFE*, 2019) so

this conformation is not unexpected. We propose that there is an equilibrium between open and closed states in both cases, and that the Ca^{2+} free state may favour the closed state, thus giving a 3-fold lower rate of scrambling in the absence of Ca^{2+} . More work is needed to elucidate the mechanisms that fine tune the equilibrium between these different conformations of the TMEM16 scramblase pathway.

7. Because scrambling by TMEM16K is 300-fold slower than the fungal homologs, it seems possible that TMEM16K is not a scramblase but rather is an ion channel and that the X-ray structure is anomalous and misleading.

We now have data on the activity of TMEM16K in shorter chain lipids, which shows scrambling at a rate similar to the fungal homologues, confirming that TMEM16K is a robust scramblase. In these conditions we could also observe non-selective ion channel activity mediated by TMEM16K. This data shows that TMEM16K behaves like other TMEM16-type scramblases in that it is a dual-function scramblase/non-selective channel. Our molecular dynamics simulation data suggests that the open-groove conformation seen in the X-ray structure enables headgroup translocation, consistent with what was reported for the fungal homologue nhTMEM16 (Bethel and Grabe, PNAS, 2016; Jiang et al., eLife, 2017; Lee et al., Nat Comms, 2018).

8. The X-ray structure shows a 4th Ca ion bound at the N-terminal end of TM10, maybe coordinated by the carbonyl oxygen of E586, but the authors do not say anything about this. This ion is missing in both cryo structures. While this site is luminal and unlikely to be physiologically relevant to Ca-dependent gating, it is possible that the difference in structure of the cryo and X-ray structures is related to binding of this ion. These issues must be explained.

The 4th Ca^{2+} ion is only seen in the higher resolution LCP X-ray structure; it is not observed in the vapour diffusion crystal structure. As the two X-ray structures have very similar conformations, we reasoned that this ion was a result of the LCP crystallisation rather than a functional feature of the open groove conformation. We have now added a sentence to describe this ion in the main results section for clarity (page 12, Lns 22-24).

9. It is also strange that CGMD shows no scrambling of the cryo-EM structure despite the fact that the experimental data show that scrambling is quite robust in the absence of Ca (Ca-free rate constant is 25% that of the Ca-stimulated system). The Ca-stimulated system shows dozens of transitions (Fig. 4C-D), while the Ca-free system does not show even a single lipid entering the groove from either side (Suppl. Fig. 8i). In that regard, it seems that the authors have been a little disingenuous in showing only a 2- μsec simulation for the Ca-free state while they show 10 μsec for the Ca-stimulated state. We have now extended the simulations of the Ca-free state to match that of the Ca-stimulated state (see Supplementary Fig. 12j). Even over 10 μs , we see no scrambling from this conformation. No scrambling is possible in the cryo-EM structure as the groove is closed and it is very difficult to see how a lipid headgroup could navigate around the closed neck of the protein. Rather than this being a scrambling state of the lipid, we believe that there must be an additional, as yet unobserved, Ca^{2+} free state, which we speculate would resemble the open groove conformation, without the Ca^{2+} ions stabilizing it, as has been observed for nhTMEM16 (Kalienkova, et al., eLife, 2019). This open groove Ca^{2+} free state could be less stable than its equivalent Ca^{2+} bound state, so it would be more inclined to collapse back into the closed groove conformation.

10. The authors suggest that the Asp615Asn mutation may explain the SCAR10 ataxia because it is involved in Ca binding which could affect channel structure or gating. This needs verification because there are no published experimental data showing that this amino acid in any TMEM16 protein affects channel function. The authors should test this mutant in the liposome assay.

We thank this reviewer for prompting us to test this. In response to this suggestion we made this mutation in both insect cell and mammalian expression systems. The purified protein has similar stability to the wildtype protein, both have melting temperatures of 54-55°C (using the Prometheus

instrument from Nanotemper to measure changes in intrinsic Trp fluorescence while heating the sample). We tested the activity of the mutant in scramblase assays and found that, at least under the conditions tested, the activity was very similar with and without the mutation. We have changed the text of the paper (Page 13, Lns 6-8) and removed the reference to this mutation in the abstract to reflect this finding. We thank the reviewer for suggesting that we check this aspect of the project.

11. The proteoliposome assay shows TMEM16K scrambles lipids at a rate 300-fold slower than the fungal TMEM16s. One explanation the authors propose to explain this difference is “sensitivity to membrane composition”. This suggests that the experiments on TMEM16K and the fungal proteins were carried out under different conditions. Because both measurements were presumably performed by the Accardi lab, this explanation should be tested in side-by-side experiments. Another explanation the authors propose is “small molecules, proteins, or lipid cofactors”. This needs better justification. Where do the authors imagine these cofactors coming from? An explanation that the authors do not offer is that the TMEM16K protein is partly denatured or inactive. In this regard, the authors should explain and show data for how many proteins are incorporated per liposome.

We apologise for the confusion. The rates shown were from the same lipid composition, from side-by-side reconstitutions of aTMEM16 and TMEM16K. We now added gels showing that both proteins reconstitute into proteoliposomes with similar efficiency (Supplementary Figure 3g), indicating that the difference in activity is not due to denaturation. Further, we now show that the activity of TMEM16K is greatly increased with shorter chain lipids which form thinner membranes (Fig. 2), suggesting that the membrane composition is indeed a key regulator of TMEM16K activity. We have removed the references to “small molecules, proteins or lipid cofactors”.

Minor issues:

- Although the TMEM16 nomenclature is favored by many investigators in the field, the ANO terminology is the preferred name approved by the nomenclature committees and should be cross-referenced.

We apologise for this omission. We added the phrase “(also known as anoctamins ANO1-10)” to the first paragraph of the main text.

- Page 5. Line 2. Should include reference #8.

We apologise for the omission and we have added this reference.

- Fig. 1f. By my eye, I cannot tell the difference between the color used for TMEM16K 0.5 mM Ca and CLC-ec1.

We have now changed the colours on the images to ensure that the colours are distinct.

- The statements about ion channel activity are disconcerting. What exactly does “inconclusive” mean? Considering my concerns expressed in #2 above, it seems necessary to have a clear understanding about whether this is a channel or not.

Using proteoliposomes formed from a 50%-50% mixture of lipids with shorter (14 carbon) and longer (16-18 carbon) chains, we were able to detect channel activity in bulk flux assays (Fig. 2h). Since we have now shown that we can obtain channel activity, we have removed the statements that we had tried but not succeeded in detecting channel activity with purified protein. We now include an endpoint assay describing this activity (page 8, Lns 13- 24).

- Figure 4c,d shows that lipids move in both directions, however, it is not clear how this happens. Do the lipids pass one another going different directions or is there a concerted alternation in the direction of conduction through the pathway?

Inspection of the data suggests that lipids are only able to proceed in single file along the hydrophobic groove, which they are able to do 3-4 at a time. Each groove can process lipids in either direction, and for the direction of flipping to switch, the groove needs to fully flip the lipids already present – i.e. lipid flipping in the opposite direction happens consecutively, not concurrently.

- Fig. 4b, the gold headgroups look grey in my copy.

Whilst gold in the original image, we agree the colour does appear slightly less clear once the figure has been compressed and converted to different file formats. We have amended the legend for clarity.

- Fig. 4e,f. In the atomistic MD, it does not seem that any of the lipids make a full translocation from one side to the other, unless I misunderstand the figure. The statement on P8 “We also observed lipid scrambling upon conversion of intermediate states from the CG simulations to an atomistic description (Fig 4d), with lipids sampling the full translocation pathway.” is ambiguous. I don’t know what the authors mean by “scrambling upon conversion of intermediate states.” The authors seem to equate scrambling with occupancy of the cleft with lipids rather than actual translocation of a single lipid from one side to the other. Please clarify.

In the atomistic simulation we observe individual lipids entering, moving along, and exiting the groove; however, in the timeframe of the simulation, we do not see an individual lipid doing all three, ie fully translocating from one leaflet to the other. We have now extended these simulations from 1.1 μ s to 2.1 μ s, and amended the figure, and have clarified this section in the text accordingly.

- Page 8. The statement “nhTMEM16 is one of the few proteins to reveal lipid scrambling, within the >3500 membrane proteins in MemProtMD” is inscrutable. Does this mean that all 3500 proteins in MemProtMD have been examined for scrambling in silico or in wet experiments? This work has been done *in silico* rather than in wet lab experiments. All 3500 membrane proteins have been studied in CGMD and are available in the MemProgMD database. The work is described in Stansfeld et al 2015¹. We have now clarified this in the text.

1. Stansfeld, P. J. *et al.* MemProtMD: Automated Insertion of Membrane Protein Structures into Explicit Lipid Membranes. *Structure* **23**, 1350–1361 (2015).

- Typo: Supplementary Figure 3d. mTMEM16A is not 5OCB.

We apologise for this error and have corrected this in the new draft.

-Page 12. These movements recapitulate the changes seen in afTMEM16 35 should also cite ref. 8. We have added the reference as suggested.

Reviewer #3:

We thank reviewer #3 for their helpful suggestions, positive comments and support for publication of the manuscript in Nat. Comms.

1. A surprising issue about the manuscript is the general neglect (or minimal discussion) about the importance of lipid role in conformational change. Considering the lipid scrambling activity and significant lipid remodeling upon conformational change makes even more unexpected not to investigate the lipid role thoroughly.

We thank the reviewer for this suggestion, which prompted us to carry out the experiments in the thinner membranes, and thus identify conditions where the activity of TMEM16K is greatly enhanced (Fig. 2b-d). Additionally, we characterized the lipid headgroup selectivity of the scramblase and found that it is poorly selective, consistent with our MD simulations, although we do observe a moderate discrimination against PS, ~3-fold (Fig. 2e-f).

2. The crystal structure has been obtained in LCP crystallization and it has been mentioned in the manuscript that there are several MAG lipids in the dimer interface; it would be interesting if there were any indication of lipid binding into the grooves.

We looked in all our structures for the presence of a lipid or at least evidence for a headgroup in the lipid headgroup translocation groove, but we did not observe any such density. This suggests that the lipid headgroups move through the groove without specific high affinity binding sites, so they are not visible in the structure. Interestingly, we did see density for detergents, Mag7.9 and one lipid molecule bound within the dimer cavity in various structures. We have now added additional panels to supplementary figures 6 and also a new supplementary fig. 9, with panels g-j showing density for lipids in the new higher resolution cryo-EM structure. These figures illustrate the lipid and detergent distributions in the dimer cavity. Since lipids were not added to the purification, this lipid binding site may represent a natural lipid binding site. It lies on the back of TM3-5 unit, potentially affecting the function of the protein. However, mutagenesis and functional studies would be required to confirm if this lipid binding site is of significance for scramblase or channel function and these experiments are beyond the scope of the present manuscript.

3. In the CG simulation, as there are several scrambling events, it would be very interesting to see if there is a preference for any of the lipid used in the simulation.

We agree that this is an interesting question. Our membranes are built of ca. 65% POPE, 32% POPC and 3% POPS. In the flipping data, we get an estimate of ca. 77% POPE, 20% POPC and 3% POPS flipped (based on 35 flipping events in total). Whilst sampling of these events is insufficient to derive significance values, it is suggestive that lipid scrambling is non-selective or minimally selective, consistent with our new experimental findings (Fig. 2e-f).

In silico scrambling analysis has shown interesting results that justifies a more detailed description:
a. In the CG MD simulation there are enough scrambling events in both directions to make a good estimation of kinetic rate of lipid scrambling by TMEM16K.

b. There is an interesting observation in the CG simulation (Fig 4c,d). In the first 2 μ s it seems that flipping is a directional process (i.e. lipids are only flipping from the cytoplasmic to luminal leaflet). Is there any explanation for this observation?

c. Presenting the same quantification of lipid flipping for the atomistic simulation (like Fig 4c,d) would be nice.

a) We can indeed obtain scrambling rates from the CG data, however, based on previous work of ours and other groups, it is likely that the rates obtained by Martini simulation are somewhat overestimated. This is an unavoidable result of the smoothed landscape of the CG force field. Therefore, we are hesitant to include quantified rates in the manuscript, except for relative comparison.

b) It is true that, for this data, the flipping strongly favours the one direction for the first 2 μ s or so. This is an observation we fail to see in repeat data (included below for reference), so is likely a random effect.

c) This would indeed be very useful, however we simply do not see enough movement in the sampling limits of the atomistic simulation (extended here to 2.1 μs). However, we can estimate that the perceived rate is ~ 10 fold slower than the CG rate, which fits with previous estimates [SJ Marrink. 2013; Chem Soc Rev; 10.1039/C3CS60093A]. We have added a section stating this to the manuscript (Page 11, Ins 8-16).

There is no description of the residues involved in the flipping process. Also a comparison with the findings of previous studies (Bethel & Grabe PNAS and Jiang et al eLife) would be beneficial.

We have now provided a description of the residues identified by our MD analysis as being involved in lipid scrambling in the manuscript text (Page 11, In 17 to Page 12, line 5) and in Fig. 4b and Fig 6e-h.

The studies Reviewer #3 cites both use MD simulations to identify residues at various sites along the nhTMEM16 scramblase groove that show preferential interactions with lipid headgroups, as evidenced by their statistically higher phosphate occupancy. Bethel & Grabe's¹ study (Bethel & Grabe, PNAS, 113, 14049–14054, 2016) focuses on external sites which distort the membrane and serve as staging areas for lipids at the groove openings. Jiang et al. (eLife, 6, 2017) perform a similar analysis and identified sites within the groove which seem to provide a path of punctuated waypoints by which a lipid is able to traverse.

Atomistic analysis of TMEM16K's scrambling shows a more diffuse mode of action, with a larger number of residues contributing to the scrambling process, but with a relatively minimal variance in phosphate occupancy. This is most likely due to the poor conservation of the nhTMEM16 charged residues highlighted by Jiang et al and Bethel & Grabe as comprising these sites in TMEM16K. Indeed, Bethel & Grabe cite this lack of conservation as potential evidence that TMEM16K is *not* a lipid scramblase – the prevailing opinion at the time.

We have amended the text to highlight this interesting result and thank the reviewer for their suggestion.

Atomistic simulations have been started after a short CG simulation rather than starting directly from the structure, is there any reason for this protocol? The electrostatic interactions are poorly described in the Martini force field; therefore, the Ca binding site probably would not be accurately preserved during the CG simulation. Also, as the atomistic simulations have been performed in a homogenous bilayer (POPC only), they do not require equilibrated mixing of different lipid species.

To clarify this section: we performed a CG simulation in a POPC membrane to allow the lipids to equilibrate and enter the hydrophobic groove. We then used the position of the CG protein to build a new molecular system using the structural-based coordinates, including the Ca^{2+} coordinates. This means that any artefacts from the Martini simulation would be avoided. This is a standard protocol we developed and use for our system-setup, see Stansfeld, et al., JCTC, (2011). We apologise that this was not made clear in the original manuscript; we have now added some detail to this section.

As the experimental results about the ion channel activity were inconclusive, it would be interesting to know if MD simulation is showing any indication about ion transport? Or whether the grooves serve as the pathway for ion conductance also (as shown for nhTMEM16, Jiang et al 2016 ref 8).

We do not observe any ion flux in the 2.1 μ s atomistic simulation data. This would normally be sufficient sampling to observe multiple ion translocation events, suggesting that this state of TMEM16K is unable to transport Na⁺ or Cl⁻. It is possible that the ion conduction conformation is not exactly the same as the conformation seen here. There may be intermediate states (as seen for other TMEM16 family members), with a groove that is partially closed, or with side chains in a different position. For example we do see Cl⁻ ions enter the luminal vestibule and remain in that region for some time, but in this conformation they're blocked from transport by Lys606. If Lys606 were to move, then we might feasibly get transport. This observation is however somewhat speculative, so we decided not to include it in the manuscript.

In Fig 6c, considering the medium resolution of the EM maps how much the side chains' positioning is reliable? Addition of actual maps makes the comparison trustworthy.

We have now obtained an improved, higher resolution cryo-EM structure in 2 mM Ca²⁺, where many of the sidechains are clearly visible. We have now added maps to show the quality of the data (Supplementary Figs. 9 and 11). We also improved the lower resolution maps using Relion 3 and by improving the pixel size estimate. We used the higher resolution cryo-EM structure, to check for any improvements that could be made to the 430 nM and EGTA structures. As we now have a higher resolution version of the closed groove structure, we feel it is justified to include this analysis.

References need a careful check and editing. For example, reference number 31, 33 and 39 (Malvezzi et al) are identical!

We apologise for this error and have corrected this in the latest version. We believe we have now removed all the duplicate references.

Reviewer #4:

We thank reviewer #4 for their positive comments and helpful suggestions on how to improve the paper. Below are responses to the individual suggestions.

1. The authors state that no currents could be recorded from cells transfected with TMEM16K, indicating that TMEM16K is exclusively a scramblase. As Figure 1 shows that TMEM16K is primarily localized to the ER and currents are being recorded from the plasma membrane, it is possible that the lack of channel activity is due to a low expression level in the plasma membrane. As the authors further state that bilayer and flux-based assays are inconclusive, it would be better to refrain from claiming that TMEM16K is not a channel without more definitive data. The scramblase activity appears to be robust and is the focus of this work.

We apologise for the confusion, we did not intend to state or imply that TMEM16K is exclusively a scramblase. We agree with the reviewer and we intended to say that it did not give PM currents, but that it could have channel activity in the ER. We could not investigate the effect on channel activity because we couldn't do the assay on the longer lipid liposomes. In membranes formed from short chain lipids, TMEM16K has similar scramblase and ion channel activity to afTMEM16, much higher than in the longer chain lipids used in the original experiments. Under similar conditions, we were able to show that TMEM16K does have nonspecific cation channel activity, in line with many other TMEM16 (Fig. 2h). So we no longer state that the ion channel experiments are inconclusive, as we have improved this data.

2. Extensive 2D classification (5-6 rounds) was used to filter the cryo-EM image data prior to further classification steps. Such extensive classification can exclude both “bad” and “good” particles. As the Ca²⁺-bound cryo-EM structure is quite distinct from the x-ray structures, is it possible that particles belonging to alternative conformations are being excluded during 2D or 3D classification. A thorough understanding of any conformational heterogeneity present in the cryo-EM data would be very helpful for starting to understand why such different conformations were obtained by X-ray crystallography and by cryo-EM. If the cryo-EM data set does indeed contain only a single conformation, a more complete discussion should be included as to why the protein adopted such distinct conformational states under similar conditions. Is the presence of the extra Ca²⁺ binding site on the TM10-TM10 axis contributing to the open conformation?

We have been on the lookout for the presence of a mixture of particles in the cryo-EM, as we hoped to see several conformations in the cryo-EM samples. Even in the higher resolution dataset, we were only able to see the closed conformation, we did not find any additional 2D and 3D classes that could be interpreted as the open-groove conformation. We added a note to the methods section to clarify this point. Admittedly our datasets have hundreds of thousands of particles, rather than millions, but nevertheless, we would have expected to see some indication of additional classes, if they were present. A recent publication on nhTMEM16 does show a mixture of different conformations (Kalienkova, et al., eLife, 8:e44364, 2019), but in our samples we only observed one conformation.

3. In Figure 5, HOLE is used to calculate the volume of the scramblase groove in the open and closed configurations. At 4.5Å resolution the side-chain densities are unlikely to be well enough resolved for accurately modelling. As differences in side-chain positions can dramatically alter the shape of a cavity, a 4.5Å structure is too low for such calculations. It would be better to remove the HOLE calculation and the side-chains from Figure 5 and focus on the changes in the positions of TM3 and TM4. A similar analysis of the Ca²⁺-free state would also be informative for the reader.

We agree with the reviewer that it was somewhat speculative to use HOLE and to show sidechains with the 4.5 Å map. As we now have an improved structure, with data to 3.5 Å and well defined sidechains, we feel that the use of HOLE and the representation of sidechains is now better justified, so we have redone these figures with the higher resolution structure. In practice, the 3.5 Å and 4.2 Å resolution structures are very similar, so this change does not affect the interpretation of the structures.

4. On a related note, how were the MD analyses of the Ca²⁺-bound closed conformation performed? Is the accuracy of the model based on a 4.5 Å structure sufficient for the simulation?

The 4.2 Å cryo-EM structure was built based on the much higher resolution X-ray structures and in practice it proved to be very similar to the 3.5 Å 2 mM Ca²⁺ structure. While 4 Å is our usual, unofficial limit for MD simulations, we are confident that the basis of the cryo-EM coordinates makes the structure suitable for MD simulations.

REVIEWERS' COMMENTS:

Reviewer #1 (Remarks to the Author):

The authors performed additional experiments as to the subcellular localization and the functional dependence on lipid chain length. These data do add to the paper, but do not address my main point which is that there is no established physiological functional role for TMEM16K. Therefore, it is still not clear to me whether any of the functional properties that are being investigated have any bearing on what this protein actually does in a cell. I think this limits the impact of this work. The hypothesis that the Ca-bound closed state represents a closed state is reasonable. The explanation that Ca²⁺ seems to be co-purified with the protein is satisfactory. Even better, the structure in the presence of 2mM Ca²⁺ alleviates any concern.

Reviewer #2 (Remarks to the Author):

The authors have done an admirable job of addressing my previous concerns. Below I have noted a few minor points that the authors may wish to consider.

1. While the colocalization of 16K with the ER markers is clear, I would like to see a higher magnification of the colocalization. In the previous version of the manuscript, the authors had a nice image showing the location of 16K in ER tubules at high mag. I think a similar image should be shown using the new markers.
2. The authors show both forward and backward rate constants for scrambling, but in the methods, they state that it is assumed that the rates are equal (page 31, line 5-6). I don't get it.
3. In SFig.8d, the amino acids should be labelled. Also, in the legend this is referred to as the third Ca site, while in the text it is referred to as the fourth Ca site.

Criss Hartzell

Reviewer #4 (Remarks to the Author):

The authors have adequately addressed all of my concerns and the manuscript is now suitable for publication in nature communications.

Reviewer #1 (Remarks to the Author):

Reviewer #1: The authors performed additional experiments as to the subcellular localization and the functional dependence on lipid chain length. These data do add to the paper, but do not address my main point which is that there is no established physiological functional role for TMEM16K. Therefore, it is still not clear to me whether any of the functional properties that are being investigated have any bearing on what this protein actually does in a cell. I think this limits the impact of this work.

Response to reviewer #1: This is a difficult question to answer. We are not aware of methods that we could use to study the effects of TMEM16K scrambling in ER membranes. Nevertheless we believe that the *in vitro* methods we use here provide valuable evidence for scrambling by TMEM16K, confirming that it is capable of scrambling lipids and acting as a channel. This activity is now very much in line with the activity of related fungal and human homologues at the plasma membrane and elsewhere, so it seems highly likely that this scrambling occurs in cells, as well as in vitro. We regard further investigation of these effects are beyond the scope of this paper.

Reviewer #1: The hypothesis that the Ca-bound closed state represents a closed state is reasonable. The explanation that Ca²⁺ seems to be co-purified with the protein is satisfactory. Even better, the structure in the presence of 2mM Ca²⁺ alleviates any concern.

Response to reviewer #1: We thank the reviewer for the positive comments.

Reviewer #2 (Remarks to the Author):

We thank Professor Criss Hartzell, for his very helpful feedback and positive comments on our work.

Reviewer #2: 1. While the colocalization of 16K with the ER markers is clear, I would like to see a higher magnification of the colocalization. In the previous version of the manuscript, the authors had a nice image showing the location of 16K in ER tubules at high mag. I think a similar image should be shown using the new markers.

Response to Reviewer #2: We think our quantitative, unbiased analysis of co-localisation offers the reader convincing evidence of colocalisation, while we feel higher magnification images would not offer further insight on co-localisation. We would therefore prefer to show the cells in their entirety. We however wish to show an expanded image to the Reviewer for inspection (Response Fig 1, please see below).

Concerning the tubular structures, these are not unequivocally detectable on visual inspection regardless of the magnification.

Response Figure 1. Evidence for TMEM16K location in ER. Representative confocal images of COS-7 cells expressing TMEM16K-TEV-His10-FLAG. Cells were stained for TMEM16K (anti-FLAG: red), the ER resident protein calnexin (CNX: green) and nuclei (DAPI: blue). In the merged panel (lower right), the degree of TMEM16K and CNX overlap is shown (yellow). Scale bars = 10 μm ; magnification: 63x. The insets in each panel represent an expanded image of the section denoted by the red rectangles.

Reviewer #2: 2. The authors show both forward and backward rate constants for scrambling, but in the methods, they state that it is assumed that the rates are equal (page 31, line 5-6). I don't get it.

Response to reviewer #2: We apologise for the lack of clarity. We have changed the text in the methods section to explain this situation: "Quantification of the scrambling rate constants of TMEM16K and afTMEM16 were determined³⁶. The fluorescence time course was fit to the following equation:

$$F_{tot}(t) = f_0(L_i^{PF}(0) + (1 - L_i^{PF}(0))e^{-\gamma t}) + \frac{(1-f_0)}{D(\alpha+\beta)} \{ \alpha(\lambda_2 + \gamma)(\lambda_1 + \alpha + \beta)e^{\lambda_1 t} + \lambda_1\beta(\lambda_2 + \alpha + \beta + \gamma)e^{\lambda_2 t} \} \quad (1)$$

Where $F_{tot}(t)$ is the total fluorescence at time t , L_i^{PF} is the fraction of NBD-labelled lipids in the inner leaflet of protein free liposomes, $\gamma = \gamma'[D]$ where γ' is the second order rate constant of dithionite reduction, $[D]$ is the dithionite concentration, f_0 is the fraction of protein-free liposomes in the sample, α and β are the forward and reverse scrambling rate constants and

$$\lambda_1 = -\frac{(\alpha+\beta+\gamma) - \sqrt{(\alpha+\beta+\gamma)^2 - 4\alpha\gamma}}{2} \quad \lambda_2 = -\frac{(\alpha+\beta+\gamma) + \sqrt{(\alpha+\beta+\gamma)^2 - 4\alpha\gamma}}{2} \quad (2, 3)$$

The free parameters of the fit are f_0 and α while L_i^{PF} and γ are experimentally determined from experiments on protein-free liposomes. In protein-free vesicles a very slow fluorescence decay is visible likely reflecting a slow leakage of dithionite into the vesicles or the spontaneous flipping of the NBD-labelled lipids. A linear fit was used to estimate the rate of this process was estimated to be $L=(5.4\pm 1.6)\cdot 10^{-5} \text{ s}^{-1}$ ($n>160$)³⁵. For TMEM16K, the leak is >2 orders of magnitude smaller than the rate constant of protein-mediated scrambling and therefore was considered negligible. All conditions were tested side by side with a control preparation of afTMEM16 in standard conditions.”

3. In SFig.8d, the amino acids should be labelled. Also, in the legend this is referred to as the third Ca site, while in the text it is referred to as the fourth Ca site.

Response to reviewer #2: We have modified the figure as suggested and changed the figure legend to read “Fourth Ca^{2+} ion located on the dimer axis between TM10s on the ER surface of the membrane, found only in the LCP crystal structure.” I hope this clarifies the situation.